# Neurotrophic factor Neuritin modulates T cell electrical and metabolic state for the balance of tolerance and immunity

Hong Yu[1,2]*[†‡], Hiroshi Nishio[1,2†§], Joseph Barbi[1,2†#], Marisa Mitchell-Flack[1,2], Paolo DA Vignali[1,2¶], Ying Zheng[1,2], Andriana Lebid[1,2], Kwang-Yu Chang[1,2**], Juan Fu[1,2], Makenzie Higgins[1], Ching-Tai Huang[2††], Xuehong Zhang[3], Zhiguang Li[3], Lee Blosser[1], Ada Tam[1], Charles Drake[2‡‡], Drew Pardoll[1,2]

[1]Bloomberg-Kimmel Institute for Cancer Immunotherapy, Immunology and Hematopoiesis Division, Department of Oncology, Johns Hopkins University School of Medicine, Baltimore, United States; [2]The Sidney Kimmel Comprehensive Cancer Center, Johns Hopkins University School of Medicine, Baltimore, United States; [3]Institute of Cancer Stem Cell, Cancer Center, Dalian Medical University, Dalian, China

**\*For correspondence:**
hyu13@jhmi.edu; yuh5@uthscsa.edu

[†]These authors contributed equally to this work

**Present address:** [‡]University of Texas Health Science Center San Antonio, Mays Cancer Center, San Antonio, United States; [§]University of Pittsburgh, Carnegie Mellon, Pittsburgh, United States; [#]Department of Obstetrics and Gynecology, Keio University School of Medicine, Tokyo, Japan; [¶]Department of Immunology, Roswell Park Comprehensive Cancer Center, Buffalo, United States; [**]National Institute of Cancer Research, National Health Research Institutes, Tainan, Taiwan; [††]Infectious Diseases, Department of Medicine, Chang Gung Memorial Hospital, Taoyuan, Taiwan; [‡‡]Division of Hematology and Oncology, Herbert Irving Comprehensive Cancer Center, Columbia University Medical Center, New York, United States

**Competing interest:** The authors declare that no competing interests exist.

## eLife Assessment

The neurotrophic factor Neuritin can moderate T-cell tolerance and immunity through both regulatory T (Treg) and effector T cells, promoting Treg cell expansion and suppression while dampening effector T cells to mediate the inflammatory response. Neuritin expression influences the membrane potential, ion channels, and nutrient transporter expression patterns of CD4+ T cells, contributing to differential metabolic states in Treg and effector T cells. These findings are **solid** and **important** for understanding immune regulation involving Treg cells and effector T cells.

**Abstract** The adaptive T cell response is accompanied by continuous rewiring of the T cell's electric and metabolic state. Ion channels and nutrient transporters integrate bioelectric and biochemical signals from the environment, setting cellular electric and metabolic states. Divergent electric and metabolic states contribute to T cell immunity or tolerance. Here, we report in mice that neuritin (*Nrn1*) contributes to tolerance development by modulating regulatory and effector T cell function. *Nrn1* expression in regulatory T cells promotes its expansion and suppression function, while expression in the T effector cell dampens its inflammatory response. *Nrn1* deficiency in mice causes dysregulation of ion channel and nutrient transporter expression in Treg and effector T cells, resulting in divergent metabolic outcomes and impacting autoimmune disease progression and recovery. These findings identify a novel immune function of the neurotrophic factor *Nrn1* in regulating the T cell metabolic state in a cell context-dependent manner and modulating the outcome of an immune response.

## Introduction

Peripheral T cell tolerance is important in restricting autoimmunity and minimizing collateral damage during active immune reactions and is achieved via diverse mechanisms, including T cell anergy, regulatory T (Treg) cell mediated suppression, and effector T (Te) cell exhaustion or deletion (*ElTanbouly and Noelle, 2021*). Upon activation, Treg and conventional T cells integrate environmental cues and

adapt their metabolism to the energetic and biosynthetic demands, leading to tolerance or immunity. Tolerized versus responsive T cells are characterized by differential metabolic states. For example, T cell anergy is associated with reduced glycolysis, whereas activated T effector cells exhibit increased glycolysis (*Buck et al., 2017*; *Geltink et al., 2018*; *Peng and Li, 2023*; *Zheng et al., 2009*). Cellular metabolic states depend on electrolyte and nutrient uptake from the microenvironment (*Chapman and Chi, 2022*; *Olenchock et al., 2017*). Ion channels and nutrient transporters, which can integrate environmental nutrient changes, affect the cellular metabolic choices and impact the T cell functional outcome (*Babst, 2020*; *Bohmwald et al., 2021*; *Ramirez et al., 2018*). Each cell's functional state would correspond with a set of ion channels and nutrient transporters supporting their underlying metabolic requirements. The mechanisms coordinating the ion channel and nutrient transporter expression changes to support the adaptive T cell functional state in the immune response microenvironment remain unclear.

*Nrn1*, also known as candidate plasticity gene 15 (CPG15), was initially discovered as a neurotrophic factor linked to the neuronal cell membrane through a glycosylphosphatidylinositol (GPI) anchor (*Nedivi et al., 1998*; *Zhou and Zhou, 2014*). It is highly conserved across species, with 96% overall homology between the murine and human protein. *Nrn1* plays multiple roles in neural development, synaptic plasticity, synaptic maturation, neuronal migration, and survival (*Cantallops et al., 2000*; *Javaherian and Cline, 2005*; *Nedivi et al., 1998*; *Putz et al., 2005*; *Zito et al., 2014*). In the immune system, *Nrn1* expression has been found in FOXP3$^+$ Treg and follicular regulatory T cells (Tfr; *Gonzalez-Figueroa et al., 2021*; *Vahl et al., 2014*), T cells from transplant tolerant recipients (*Lim et al., 2013*), anergized CD8 cells or CD8 cells from tumor-infiltrating lymphocytes in mouse tumor models (*Schietinger et al., 2012*; *Schietinger et al., 2016*; *Singer et al., 2016*), and in human Treg infiltrating breast cancer tumor tissue (*Plitas et al., 2016*). Soluble *Nrn1* can be released from Tfr cells and act directly on B cells to suppress autoantibody development against tissue-specific antigens (*Gonzalez-Figueroa et al., 2021*). Despite the observation of *Nrn1* expression in Treg cells and T cells from tolerant environments (*Gonzalez-Figueroa et al., 2021*; *Lim et al., 2013*; *Plitas et al., 2016*; *Schietinger et al., 2012*; *Schietinger et al., 2016*; *Singer et al., 2016*), the roles of *Nrn1* in T cell tolerance development and Treg cell function have not been explored, and the functional mechanism of *Nrn1* remains elusive. This study demonstrates in mice that the neurotrophic factor *Nrn1* can moderate T cell tolerance and immunity through both Treg and Te cells, impacting Treg cell expansion and suppression while controlling inflammatory response in Te cells.

## Results

### *Nrn1* expression and function in T cell anergy

To explore the molecular mechanisms underlying peripheral tolerance development, we utilized a system we previously developed to identify tolerance-associated genes (*Huang et al., 2004*). We compared the gene expression patterns associated with either a T effector/memory response or tolerance induction triggered by the same antigen but under divergent in vivo conditions (*Huang et al., 2004*). Influenza hemagglutinin (HA) antigen-specific TCR transgenic CD4 T cells were adoptively transferred into WT recipients with subsequent HA-Vaccinia virus (VacHA) infection to generate T effector/memory cells while tolerogenic HA-specific CD4s were generated by transfer into hosts with transgenic expression of HA as self-antigen (C3-HA mice, *Figure 1A*.; *Huang et al., 2004*). One of the most differentially expressed genes upregulated in the anergy-inducing condition was *Nrn1*. *Nrn1* expression was significantly higher among cells recovered from C3-HA hosts vs. cells from VacHA infected mice at all time points tested by qRT-PCR (*Figure 1A*). To further confirm the association of *Nrn1* expression with T cell anergy, we assessed *Nrn1* expression in naturally occurring anergic polyclonal CD4$^+$ T cells (Ta), which can be identified by surface co-expression of Folate Receptor 4 (FR4) and the ecto-5′-nucleotidase CD73 (Ta, CD4$^+$CD44$^+$FR4$^{hi}$CD73$^{hi}$ cells; *Kalekar et al., 2016*). *Nrn1* expression was significantly higher in Ta than in naïve CD4 (Tn, CD4$^+$CD62L$^+$CD44$^-$FR4$^-$CD73$^-$) and antigen-experienced cells (Te, CD4$^+$CD44$^+$FR4$^-$CD73$^-$) under steady-state conditions measured by both qRT-PCR and western blot (*Figure 1B*, *Figure 1—source data 1*). Given that Treg cells, like anergic cells, have roles in maintaining immune tolerance, we queried whether *Nrn1* is also expressed in Treg cells. *Nrn1* expression can be detected in nTreg and induced Treg (iTreg) cells generated in vitro (*Figure 1C*, *Figure 1—source data 3 and 4*).

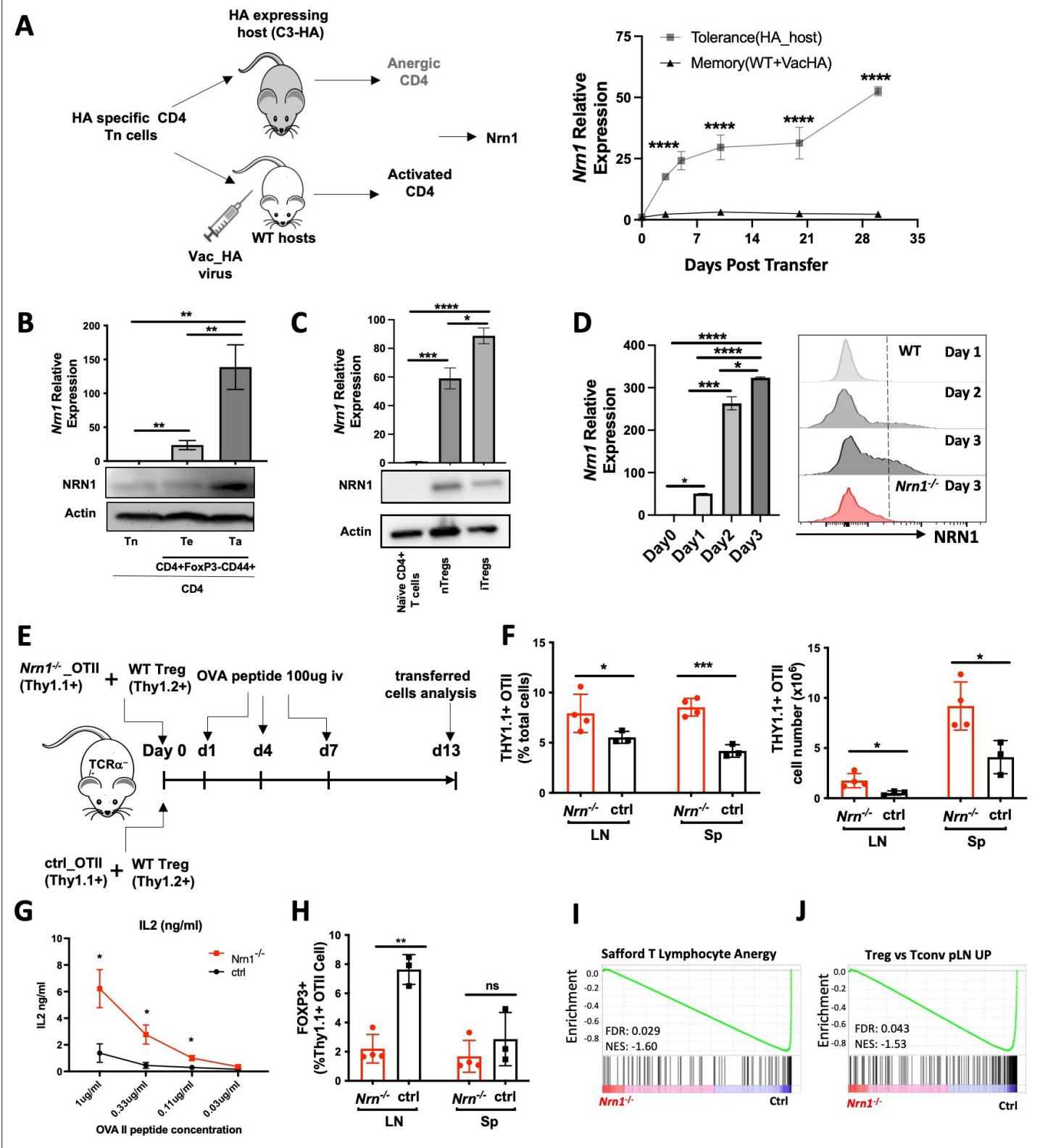

**Figure 1.** *Nrn1* expression and function in anergic T cells. (**A**) Experimental scheme identifying Nrn1 in anergic T cells and qRT-PCR confirmation of *Nrn1* expression in HA-specific CD4 cells recovered from HA-expressing host vs WT host activated with Vac_HA virus. (**B**) qRT-PCR and western blot detecting *Nrn1* expression in naïve CD4+CD62LhiCD44lo Tn cell, CD4 effector CD4+FOXP3-CD44hiCD73-FR- Te cells and CD4 anergic CD4+FOXP3-CD44hiCD73+FR+ Ta cells. (**C**) *Nrn1* expression was measured by qRT-PCR and western blot among naive CD4+ T cells, CD4+FOXP3+ nTreg, and in vitro

*Figure 1 continued on next page*

*Figure 1 continued*

generated iTregs. (**D**) *Nrn1* expression was detected by qRT-PCR and flow cytometry among WT naïve CD4[+] cells and activated CD4[+] cells on days 1, 2, and 3 after activation. *Nrn1*[-/-] CD4 cells were also stained for NRN1 3 days after activation. qPCR Data are presented as average ± SEM. *p<0.05, **p<0.01, ***p<0.001, ****p<0.0001. Triplicates were used. Ordinary one-way ANOVA was performed for multi-comparison. (**E–J**). Anergy induction in vivo. (**E**) Experimental outline evaluating anergy development in vivo: 2x10[6] Thy1.1[+] *Nrn1*[-/-] or ctrl CD4 OTII T cells were co-transferred with 5x10[5] Thy1.2[+]Thy1.1[-] WT Treg cells into TCRα[-/-]mice. Cells were recovered on day 13 post-transfer. (**F**) Proportions and numbers of OTII cells recovered from recipient spleen; (**G**) IL2 secretion from OTII cells upon ex vivo stimulation with OVA peptide. (**H**) FOXP3[+] cell proportion among Thy1.1[+] *Nrn1*[-/-] or ctrl CD4 cells. (**I & J**) *Nrn1*[-/-] vs ctrl OTII cells recovered from the peptide-induced anergy model were subjected to bulk RNASeq analysis. GSEA comparing the expression of signature genes for anergy (**I**) and Treg (**J**) among ctrl and *Nrn1*[-/-] OTII cells. Data are presented as mean ± SEM and representative of three independent experiments (N≥4 mice per group). *p<0.05, **p<0.01, ***p<0.001. Unpaired Student's t-tests were performed.

The online version of this article includes the following source data and figure supplement(s) for figure 1:

**Source data 1.** PDF file containing the Figure original western blot for *Figure 1B*, indicating the relevant bands and cell types.

**Source data 2.** Original files for western blot analysis displayed in *Figure 1B*.

**Source data 3.** PDF file containing the original western blot for *Figure 1C*, indicating the relevant bands and cell types.

**Source data 4.** Original files for western blot analysis displayed in *Figure 1C*.

**Figure supplement 1.** *Nrn1* expression in T cells from tumor environment and during early T cell activation.

**Figure supplement 2.** *Nrn1*[-/-] mice body weight and immune cell profile analysis compared to *Nrn1*[+/-], and WT mice.

**Figure supplement 3.** Compromised T cell activation in *Nrn1*[-/-] cells.

To evaluate *Nrn1* expression under pathological tolerant conditions (*Cuenca et al., 2003*), we evaluated *Nrn1* expression in T cells within the tumor microenvironment. *Nrn1* expression in murine Treg cells and non-Treg CD4[+] cells from tumor infiltrates were compared to the Treg cells and non-Treg CD4[+] T cells isolated from peripheral blood. *Nrn1* mRNA level was significantly increased among tumor-associated Treg cells and non-Treg CD4 cells compared to cells from peripheral blood (*Figure 1—figure supplement 1A*). Consistent with our findings in the mouse tumor setting, the Treg and non-Treg T cells from human breast cancer infiltrates reveal significantly higher *Nrn1* expression compared to the peripheral blood Treg and non-Treg cells (*Figure 1—figure supplement 1B*; *Plitas et al., 2016*).

CD4[+] T cells may pass through an effector stage after activation before reaching an anergic state (*Adler et al., 1998*; *Chen et al., 2004*; *Huang et al., 2003*; *Opejin et al., 2020*). To evaluate the potential role of *Nrn1* expression in T cell tolerance development, we further examined *Nrn1* expression kinetics after T cell activation. *Nrn1* expression was significantly induced after CD4[+] T cell activation (*Figure 1D*). Using an NRN1-specific, monoclonal antibody, NRN1 can be detected on activated CD4[+] and CD8[+] cells (*Figure 1D*, *Figure 1—figure supplement 1C*). The significant enhancement of *Nrn1* expression after T cell activation suggests that *Nrn1* may contribute to the process of T cell tolerance development and/or maintenance. Although Treg cells express *Nrn1*, we were not able to consistently detect substantial cell surface NRN1 expression (*Figure 1D*, *Figure 1—figure supplement 1D*), likely due to NRN1 being produced in a soluble form or cleaved from the cell membrane (*Gonzalez-Figueroa et al., 2021*).

To understand the functional implication of *Nrn1* expression in immune tolerance, we analyzed *Nrn1*-deficient (*Nrn1*[-/-]) mice (*Fujino et al., 2011*). In the first evaluation of the *Nrn1*[-/-] colony, *Nrn1*[-/-] mice had consistently reduced body weight compared to heterozygous *Nrn1*[+/-], WT (*Nrn1*[+/+]) mice (*Figure 1—figure supplement 2A*). The lymphoid tissues of *Nrn1*[-/-] mice were comparable to their *Nrn1*[+/-], WT counterparts except for a slight reduction in cell number that was observed in the spleens of *Nrn1*[-/-] mice, likely due to their smaller size (*Figure 1—figure supplement 2B*). Analysis of thymocytes revealed no defect in T cell development (*Figure 1—figure supplement 2C*), and a flow cytometric survey of the major immune cell populations in the peripheral lymphoid tissue of these mice revealed similar proportions of CD4, CD8 T cells, B cells, monocytes and dendritic cells (DCs; *Figure 1—figure supplement 2D*). Similarly, no differences were found between the proportions of anergic and Treg cells in *Nrn1*[-/-], *Nrn1*[+/-], WT mice (*Figure 1—figure supplement 2E*, F), suggesting that *Nrn1* deficiency does not significantly affect anergic and Treg cell balance under steady state. Additionally, histopathology assessment of lung, heart, liver, kidney, intestine, and spleen harvested from 13 months old *Nrn1*[-/-] and *Nrn1*[+/-] did not reveal any evidence of autoimmunity (data not shown). The comparable level of anergic and Treg cell population among *Nrn1*[-/-], *Nrn1*[+/-], WT mice and lack

of autoimmunity in *Nrn1*$^{-/-}$ aged mice suggest that *Nrn1* deficiency is not associated with baseline immune abnormalities or overt dysfunction. Due to the similarity between *Nrn1*$^{+/-}$, WT mice, we have used either *Nrn1*$^{+/-}$ or WT mice as our control depending on mice availability and referred to both as 'ctrl' in the subsequent discussion.

To evaluate the relevance of *Nrn1* in CD4$^+$ T cell tolerance development, we employed the classic peptide-induced T cell anergy model (*Vanasek et al., 2006*). Specifically, we crossed OVA antigen-specific TCR transgenic OTII mice onto the *Nrn1*$^{-/-}$ background. *Nrn1*$^{-/-}$_OTII$^+$ or control_OTII$^+$ (ctrl_OTII$^+$) cells marked with Thy1.1$^+$ congenic marker (Thy1.1$^+$Thy1.2$^-$), were co-transferred with polyclonal WT Tregs (marked as Thy1.1$^-$thy1.2$^+$), into TCRa knockout mice (*Tcra*$^{-/-}$), followed by injection of soluble OVA peptide to induce clonal anergy (*Figure 1E*; *Chappert and Schwartz, 2010*; *Martinez et al., 2012*; *Mercadante and Lorenz, 2016*; *Shin et al., 2014*). On day 13 after cell transfer, the proportion and number of OTII cells increased in the *Nrn1*$^{-/-}$_OTII compared to the ctrl_OTII hosts (*Figure 1F*). Moreover, *Nrn1*$^{-/-}$_OTII cells produced increased IL2 than ctrl_OTII upon restimulation (*Figure 1G*). Anergic CD4 Tconv cells can transdifferentiate into FOXP3$^+$ pTreg cells in vivo (*Kaleka and Daniel L Mueller, 2017*; *Kalekar et al., 2016*; *Kuczma et al., 2021*). Consistent with reduced anergy induction, the proportion of FOXP3$^+$ pTreg among *Nrn1*$^{-/-}$_OTII was significantly reduced (*Figure 1H*). In parallel with the phenotypic analysis, we compared gene expression between *Nrn1*$^{-/-}$_OTII and ctrl_OTII cells by RNA Sequencing (RNASeq). Gene set enrichment analysis (GSEA) revealed that the gene set on T cell anergy was enriched in ctrl relative to *Nrn1*$^{-/-}$_OTII cells (*Figure 1I*; *Safford et al., 2005*). Also, consistent with the decreased transdifferentiation to FOXP3$^+$ cells, the Treg signature gene set was prominently reduced in *Nrn1*$^{-/-}$_OTII cells relative to the ctrl (*Figure 1J*). Anergic T cells are characterized by inhibition of proliferation and compromised effector cytokines such as IL2 production (*Choi and Schwartz, 2007*). The increased cell expansion and cytokine production in *Nrn1*$^{-/-}$_OTII cells and the reduced expression of anergic and Treg signature genes all support the notion that *Nrn1* is involved in T cell anergy development.

Anergic T cells are developed after encountering antigen, passing through a brief effector stage, and reaching an anergic state (*Chappert and Schwartz, 2010*; *Huang et al., 2003*; *Silva Morales and Mueller, 2018*; *Zha et al., 2006*). Enhanced T cell activation, defective Treg cell conversion or expansion, and heightened T effector cell response may all contribute to defects in T cell anergy induction and/or maintenance (*Chappert and Schwartz, 2010*; *Huang et al., 2003*; *Kalekar et al., 2016*; *Silva Morales and Mueller, 2018*; *Zha et al., 2006*). We first examined early T cell activation to understand the underlying cause of defective anergy development in *Nrn1*$^{-/-}$ cells. *Nrn1*$^{-/-}$ CD4$^+$ cells showed reduced T cell activation, as evidenced by reduced CellTrace violet dye (CTV) dilution, activation marker expression, and Ca$^{++}$ entry after TCR stimulation (*Figure 1—figure supplement 3A, B, C*). The reduced early T cell activation observed in *Nrn1*$^{-/-}$ CD4 cells suggests that the compromised anergy development in *Nrn1*$^{-/-}$_OTII cells was not caused by enhanced early T cell activation. The defective pTreg generation and/or enhanced effector T cell response may contribute to compromised anergy development.

## Compromised Treg expansion and suppression in the absence of *Nrn1*

The significant reduction of FOXP3$^+$ pTreg among *Nrn1*$^{-/-}$_OTII cells could be caused by the diminished conversion of FOXP3$^-$ Tconv cells to pTreg and/or diminished Treg cell expansion and persistence. To understand the cause of pTreg reduction in *Nrn1*$^{-/-}$_OTII cells (*Figure 1H*), we turned to the induced Treg (iTreg) differentiation system to evaluate the capability of FOXP3$^+$ Treg development and expansion in *Nrn1*$^{-/-}$ cells. Similar proportions of FOXP3$^+$ cells were observed in *Nrn1*$^{-/-}$ and ctrl cells under the iTreg culture condition (*Figure 2A*), suggesting that *Nrn1* deficiency does not significantly impact FOXP3$^+$ cell differentiation. To examine the capacity of iTreg expansion, *Nrn1*$^{-/-}$ and ctrl iTreg cells were restimulated with anti-CD3, and we found reduced live cells over time in *Nrn1*$^{-/-}$ iTreg compared to the ctrl (*Figure 2B*). The reduced live cell number in *Nrn1*$^{-/-}$ was accompanied by reduced Ki67 expression (*Figure 2C*). Although *Nrn1*$^{-/-}$ iTregs retained a higher proportion of FOXP3$^+$ cells 3 days after restimulation, however, when taking into account the total number of live cells, the actual number of live FOXP3$^+$ cells was reduced in *Nrn1*$^{-/-}$ (*Figure 2D*). Treg cells are not stable and are prone to losing FOXP3 expression after extended proliferation (*Feng et al., 2014*; *Floess et al., 2007*; *Li et al., 2014*; *Zheng et al., 2010*). The increased proportion of FOXP3$^+$ cells was consistent with reduced

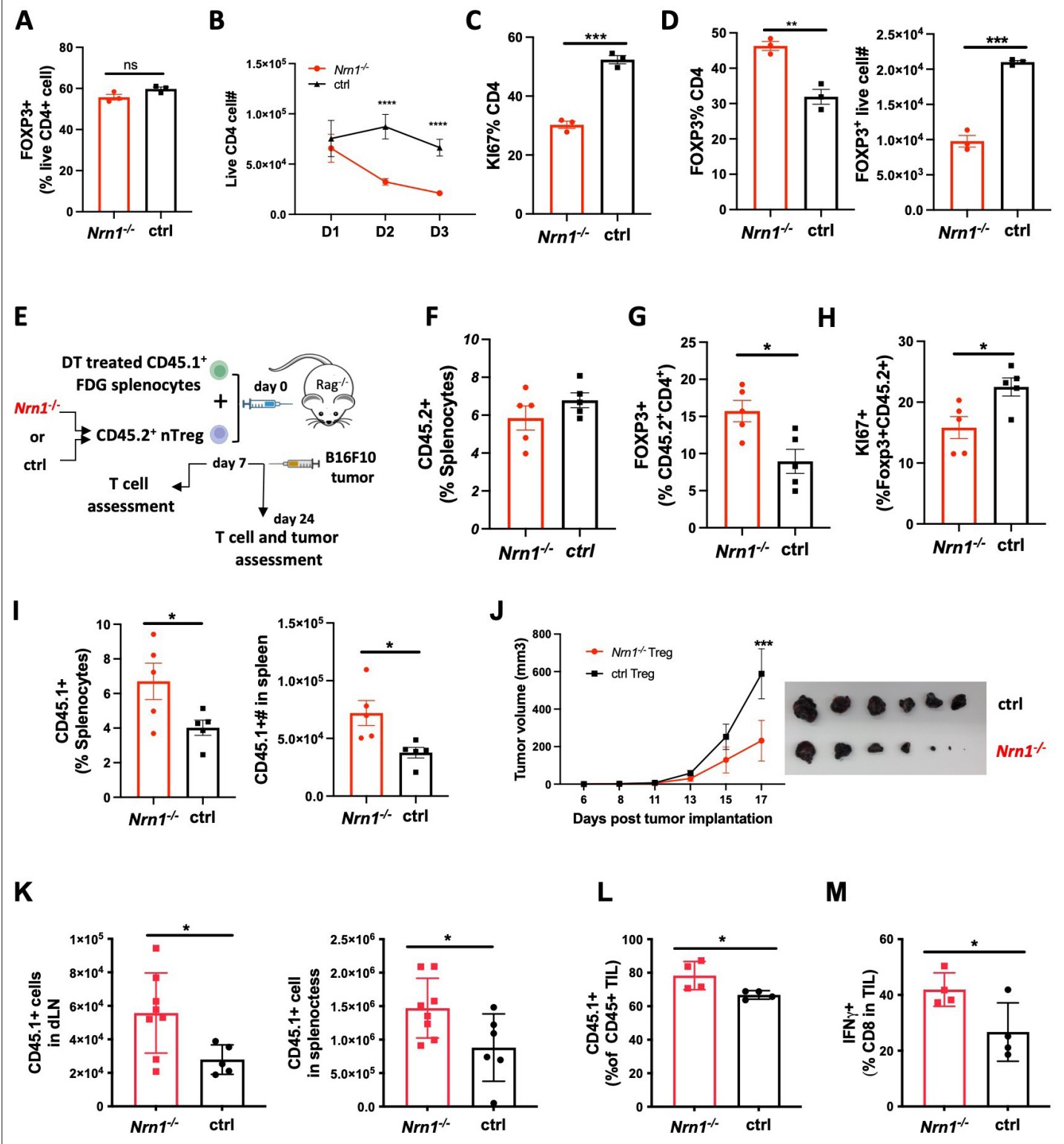

**Figure 2.** Reduced proliferation and suppression function in *Nrn1*⁻/⁻ Treg cells. (**A**) Proportion of FOXP3⁺ cells 3 days after in vitro iTreg differentiation. (**B–D**) iTreg cell expansion after restimulation. (**B**) The number of live cells from day 1 to day 3 after iTreg cell restimulation with anti-CD3. (**C**) Ki67 expression among CD4⁺FOXP3⁺ cells day 3 after restimulation. (**D**) FOXP3⁺ cell proportion and number among live CD4⁺ cells day 3 after restimulation. Triplicates in each experiment, data represent one of four independent experiments. (**E–M**) *Nrn1*⁻/⁻ or ctrl nTreg cells expansion and suppression in vivo. (**E**) The experimental scheme. CD45.2⁺ nTreg T cells from *Nrn1*⁻/⁻ or ctrl were transferred with CD45.1⁺ FDG splenocytes devoid of Tregs into the *Rag2*⁻/⁻ host. Treg cell expansion and suppression toward FDG CD45.1⁺ responder cells were evaluated on day 7 post cell transfer. Alternatively, B16F10

*Figure 2 continued on next page*

*Figure 2 continued*

tumor cells were inoculated on day 7 after cell transfer and monitored for tumor growth. (**F–J**) CD45.2$^+$ cell proportion (**F**), FOXP3 retention (**G**), and *Ki67* expression among FOXP3$^+$ cells (**H**) at day 7 post cell transfer. (**I**) CD45.1$^+$ cell proportion and number in the spleen of *Nrn1$^{-/-}$* or ctrl Treg hosts day 7 post cell transfer. (**J–L**) Treg cell suppression toward anti-tumor response. (**J**) Tumor growth curve and tumor size at harvest from *Nrn1$^{-/-}$* or ctrl nTreg hosts. (**K**) CD45.1$^+$ cell count in tumor draining lymph node (LN) and spleen. (**L**) the proportion of CD45.1$^+$ cells among CD45$^+$ tumor lymphocyte infiltrates (TILs). (**M**) IFNγ% among CD8$^+$ T cells in TILs. n≥5 mice per group. (**F–I**) represents three independent experiments, (**J–M**) represents two independent experiments. Data are presented as mean ± SEM *p<0.05, **p<0.01, ***p<0.001, ****p<0.0001. Unpaired Student's t-tests were performed.

proliferation observed in *Nrn1$^{-/-}$* cells. Thus, *Nrn1* deficiency can lead to reduced iTreg cell proliferation and persistence in vitro.

The defects observed in iTreg cell expansion in vitro prompt further examination of *Nrn1$^{-/-}$* nTreg expansion and suppression function in vivo. To this end, we tested the suppression capacity of congenically marked (CD45.1$^-$CD45.2$^+$) *Nrn1$^{-/-}$* or ctrl nTreg toward CD45.1$^+$CD45.2$^-$ responder cells in *Rag2$^{-/-}$* mice (***Figure 2E***). The CD45.1$^+$CD45.2$^-$ responder cells devoid of Treg cells were splenocytes derived from FOXP3DTRGFP (FDG) mice pretreated with diphtheria toxin (DT; ***Kim et al., 2007***; ***Workman et al., 2011***). DT treatment caused the deletion of Treg cells in FDG mice (***Kim et al., 2007***). Although the CD45.1$^-$CD45.2$^+$ *Nrn1$^{-/-}$* and ctrl cell proportions were not significantly different among hosts splenocytes at day 7 post transfer (***Figure 2F***), *Nrn1$^{-/-}$* cells retained a higher FOXP3$^+$ cell proportion and reduced Ki67 expression comparing to the ctrl (***Figure 2G and H***). These findings were similar to our observation of iTreg cells in vitro (***Figure 2C and D***). *Nrn1$^{-/-}$* Tregs also showed reduced suppression toward CD45.1$^+$ responder cells, evidenced by increased CD45.1$^+$ proportion and cell number in host splenocytes (***Figure 2I***).

To evaluate the functional implication of *Nrn1$^{-/-}$* Treg suppression in disease settings, we challenged the *Rag2$^{-/-}$* hosts with the poorly immunogenic B16F10 tumor (***Figure 2E***). Tumors grew much slower in *Nrn1$^{-/-}$* Treg recipients than those reconstituted with ctrl Tregs (***Figure 2J***). Moreover, the number of CD45.1$^+$ cells in tumor-draining lymph nodes and spleens increased significantly in *Nrn1$^{-/-}$* Treg hosts compared to the ctrl group (***Figure 2K***). Consistently, the CD45.1$^+$ responder cell proportion among tumor lymphocyte infiltrates (TILs) was also increased (***Figure 2L***), accompanied by an increased proportion of IFNγ + cells among CD8 TILs from *Nrn1$^{-/-}$* Treg hosts (***Figure 2M***). The increased expansion of CD45.1$^+$ responder cells and reduced tumor growth further confirmed the reduced suppressive capacity of *Nrn1$^{-/-}$* Treg cells.

### *Nrn1* impacts Treg cell electrical and metabolic state

To understand the molecular mechanisms associated with *Nrn1$^{-/-}$* Treg cells, we compared gene expression between *Nrn1$^{-/-}$* and ctrl iTregs under resting (IL2 only) and activation (aCD3 and IL2) conditions by RNASeq. GSEA on gene ontology database and clustering of enriched gene sets by Cytoscape identified three clusters enriched in resting *Nrn1$^{-/-}$* iTreg (***Figure 3A***, ***Figure 3—source data 1***; ***Shannon et al., 2003***; ***Subramanian et al., 2005***). The 'neurotransmitter involved in membrane potential (MP)' and 'sodium transport' clusters involved gene sets on the ion transport and cell MP regulation (***Figure 3A***, ***Figure 3—source data 1***). MP is the difference in electric charge between the interior and the exterior of the cell membrane (***Abdul Kadir et al., 2018***; ***Blackiston et al., 2009***; ***Ma et al., 2017***). Ion channels and transporters for Na$^+$ and other ions such as K$^+$, Cl$^-$ et al. maintain the ion balance and contribute to cell MP (***Blackiston et al., 2009***). MP change can impact cell plasma membrane lipid dynamics and affect receptor kinase activity (***Zhou et al., 2015***). The enrichment of 'receptor protein kinase' gene set clusters may reflect changes caused by MP (***Figure 3A***, ***Figure 3—source data 1***). Gene set cluster analysis on activated iTreg cells also revealed the enrichment of the 'ion channel and receptor' cluster in *Nrn1$^{-/-}$* cells (***Figure 3B***, ***Figure 3—source data 2***), supporting the potential role of *Nrn1* in modulating ion balances and MP.

The 'Neurotransmitter receptor activity involved in regulation of postsynaptic membrane potential' gene set was significantly enriched under resting and activation conditions In *Nrn1$^{-/-}$* cells (***Figure 3C and D***; ***Figure 3—source data 3***). The α-amino-3-hydroxy-5-methyl-4-isoxazolepropionic acid receptor (AMPAR) subunits *Gria2* and *Gria3* are the major components of this gene set and showed increased expression in *Nrn1$^{-/-}$* cells (***Figure 3D***). AMPAR is an ionotropic glutamate receptor that mediates fast excitatory synaptic transmission in neurons. *Nrn1* has been reported as an accessory protein for AMPAR (***Pandya et al., 2018***; ***Schwenk et al., 2012***; ***Subramanian et al., 2019***), although

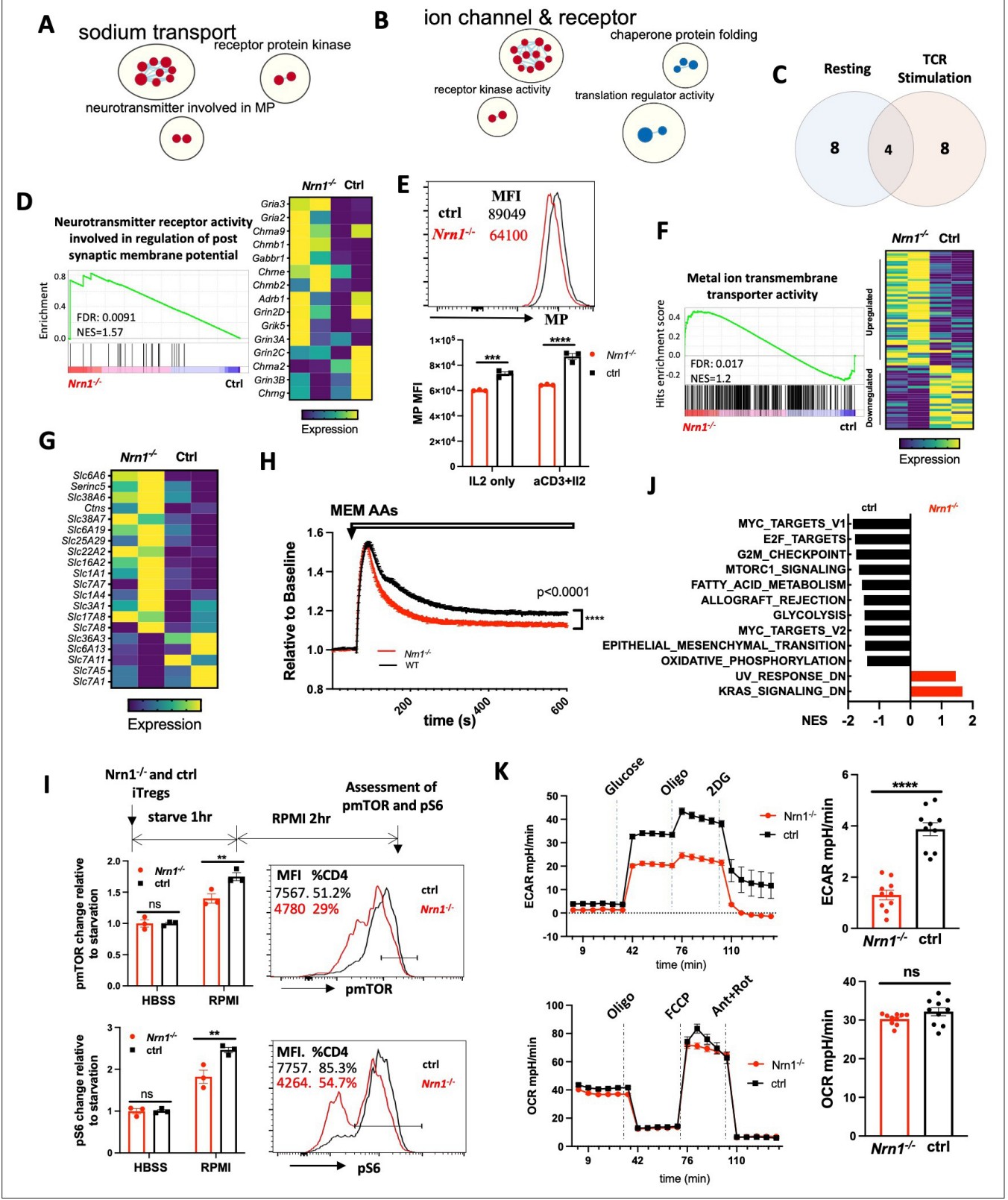

**Figure 3.** *Nrn1* expression impacts Treg cell electrical and metabolic state. (**A–C**). Gene sets clusters enriched in *Nrn1⁻/⁻* and ctrl iTreg cells. Gene sets cluster analysis via Cytoscape was performed on Gene ontology Molecular Function (GO_MF) gene sets. The results cutoff: pvalue <0.05 and FDR q-value ≤0.1. (**A**) Gene sets cluster in *Nrn1⁻/⁻* iTreg cells cultured under resting conditions (IL2 only; *Figure 3—source data 1*). (**B**) Gene sets clusters in *Nrn1⁻/⁻* and ctrl iTreg cells reactivated with anti-CD3 (*Figure 3—source data 2*). (**C**) Comparison of enriched gene sets in *Nrn1⁻/⁻* under resting vs.

*Figure 3 continued on next page*

*Figure 3 continued*

activating condition (*Figure 3—source data 3*). (**D–F**) Changes relating to cell electric state. (**D**) Enrichment of 'GOMF_Neurotransmitter receptor activity involved in the regulation of postsynaptic membrane potential' gene set and enriched gene expression heatmap. (**E**) Membrane potential was measured in *Nrn1*[-/-] and ctrl iTreg cells cultured in IL2 or activated with anti-CD3 in the presence of IL2. Data represent three independent experiments. (**F**) Enrichment of 'GOMF_Metal ion transmembrane transporter activity' gene set and enriched gene expression heatmap (*Figure 3—figure supplement 1A*). (**G–K**) Metabolic changes associated with *Nrn1*[-/-] iTreg. (**G**) Heatmap of differentially expressed amino acid (AA) transport-related genes (from 'MF_Amino acid transmembrane transporter activity' gene list) in *Nrn1*[-/-] and ctrl iTreg cells. (**H**) AAs induced MP changes in *Nrn1*[-/-] and ctrl iTreg cells. Data represent three independent experiments. (**I**) Measurement of pmTOR and pS6 in iTreg cells that were deprived of nutrients for 1 hr and refed with RPMI for 2 hr. (**J**) Hallmark gene sets significantly enriched in *Nrn1*[-/-] and ctrl iTreg. NOM p-val <0.05, FDR q-val <0.25. (**K**) Seahorse analysis of extracellular acidification rate (ECAR) and oxygen consumption rate (OCR) in *Nrn1*[-/-] and ctrl iTreg cells. n=6–10 technical replicates per group. Data represent three independent experiments. **p<0.01, ***p<0.001, ****p<0.0001. Unpaired student t-test for two-group comparison. Unpaired t-test (**H**, **K**), two-way ANOVA (**E**, **I**). ns, not significant.

The online version of this article includes the following source data and figure supplement(s) for figure 3:

**Source data 1.** Gene sets enriched in *Nrn1*[-/-] iTreg cells cultured under the resting condition.

**Source data 2.** Gene sets enriched in *Nrn1*[-/-] iTreg cells cultured under the reactivating condition.

**Source data 3.** Comparison of gene sets enriched in *Nrn1*[-/-] iTreg cells cultured under the resting and TCR restimulation conditions.

**Figure supplement 1.** Heatmap of differentially expressed genes and Hallmark gene set enrichment.

**Figure supplement 2.** Characterization of *Nrn1*[-/-] naïve CD4 T cells and effect of NRN1 blockade on WT iTreg cell differentiation and expansion.

the functional implication of *Nrn1* as an AMPAR accessory protein remains unclear. The enrichment of MP related gene set prompted the examination of electric status, including MP level and ion channel expressions. We examined the relative MP level by FLIPR MP dye, a lipophilic dye able to cross the plasma membrane, which has been routinely used to measure cell MP changes (*Dvorak et al., 2021*; *Joesch et al., 2008*; *Nik et al., 2017*; *Whiteaker et al., 2001*). When the cells are depolarized, the dye enters the cells, causing an increase in fluorescence signal. Conversely, cellular hyperpolarization results in dye exit and decreased fluorescence. Compared to ctrl iTreg cells, *Nrn1*[-/-] exhibits significant hyperpolarization under both resting and activation conditions (*Figure 3E*). Consistent with the MP change, the 'MF_metal ion transmembrane transporter activity' gene set, which contains 436 ion channel related genes, was significantly enriched and showed a different expression pattern in *Nrn1*[-/-] iTregs (*Figure 3F*; *Figure 3—figure supplement 1A and B*). The changes in cellular MP and differential expression of ion channel and transporter genes in *Nrn1*[-/-] implicate the role of *Nrn1* in the balance of electric state in the iTreg cell.

MP changes have been associated with changes in amino acid (AA) transporter expression and nutrient acquisition, which in turn influences cellular metabolic and functional state (*Yu et al., 2022*). To understand whether MP changes in *Nrn1*[-/-] are associated with changes in nutrient acquisition and thus the metabolic state, we surveyed AA transport-related gene expression using the 'Amino acid transmembrane transporter activity' gene set and found differential AA transporter gene expression between *Nrn1*[-/-] and ctrl iTregs (*Figure 3G*). Upon AA entry through transporters, the electric charge carried by these molecules may transiently affect cell membrane potential. Differential AA transporter expression patterns may have different impacts on cellular MP upon AA entry. Thus, we loaded *Nrn1*[-/-] and ctrl iTreg with FLIPR MP dye in the HBSS medium and tested cellular MP change upon exposure to MEM AAs. The AA-induced cellular MP change was reduced in *Nrn1*[-/-] compared to the ctrl, reflective of differential AA transporter expression patterns (*Figure 3H*). Electrolytes and AAs entry are critical regulators of mTORC1 activation and T cell metabolism (*Liu and Sabatini, 2020*; *Saravia et al., 2020*; *Sinclair et al., 2013*; *Wang et al., 2020*). We examined mTORC1 activation at the protein level by evaluating mTOR and S6 phosphorylation via flow cytometry. We found reduced phosphorylation of mTOR and S6 in activated *Nrn1*[-/-] iTreg cells (*Figure 3—figure supplement 1C*). We further performed a nutrient-sensing assay to evaluate the role of ion and nutrient entry in mTORC1 activation. *Nrn1*[-/-] and ctrl iTreg cells were starved for one hour in a nutrient-free buffer, followed by adding RPMI medium with complete ions and nutrients, and cultured for two more hours. While adding the medium with nutrients clearly increased the mTOR and S6 phosphorylation, the degree of change was significantly less in *Nrn1*[-/-] than in the ctrl (*Figure 3I*). Consistently, GSEA on Hallmark gene sets reveal reduced gene set enrichment relating to the mTORC1 signaling, corroborating the reduced pmTOR and pS6 detection in *Nrn1*[-/-] cells. Moreover, *Nrn1*[-/-] cells also showed reduced expression of glycolysis, fatty acid metabolism, and oxidative phosphorylation related gene sets under both resting and activating

conditions (*Figure 3J*, *Figure 3—figure supplement 1D*), indicating changes in metabolic status. Since previous work has identified mTORC1 to be an important regulator of aerobic glycolysis and given that our GSEA data suggested changes in glycolysis (*Figure 3J*; *Salmond, 2018*), we performed the seahorse assay and confirmed reduced glycolysis among *Nrn1*[-/-] cells (*Figure 3K*). Examination of mitochondrial bioenergetic function revealed a similar oxygen consumption rate (OCR) between *Nrn1*[-/-] and ctrl cells (*Figure 3K*). Thus, *Nrn1* expression can affect the iTreg electric state, influence ion channel and nutrient transporter expression, impact nutrient sensing, modulate metabolic state, and contribute to Treg expansion and suppression function.

We have observed significant changes in the electrical and metabolic state among *Nrn1*[-/-] iTreg compared to the ctrl. Because *Nrn1* can be expressed on the cell surface, one question arises whether the changes observed in *Nrn1*[-/-] cells were caused by the functional deficiency of *Nrn1* or arose secondary to potential changes in cell membrane structure originating at the *Nrn1*[-/-] naive T cell stage. To answer this question, we first examined potential changes in electrical and metabolic status among *Nrn1*[-/-] naive CD4 T cells. The *Nrn1*[-/-] naïve CD4 T cells showed similar resting MP and AA-induced MP changes compared to the ctrl cells (*Figure 3—figure supplement 2A, B*). We also observed comparable glycolysis and mitochondrial bioenergetic function between *Nrn1*[-/-] naïve CD4 T cells and the ctrl (*Figure 3—figure supplement 2C*). These results suggest the electrical and metabolic state in *Nrn1*[-/-] T cells are comparable to the ctrl cells at the naive cell stage. To further rule out the possibility that the observed changes in *Nrn1*[-/-] iTreg are secondary to developmental structural changes, not *Nrn1* functional deficiency, we differentiated WT T cells in the presence of antagonistic NRN1 antibody and compared to the WT ctrl and *Nrn1*[-/-] iTreg cells. WT iTreg cells differentiated in the presence of *Nrn1* antibody exhibit reduced resting MP, similar to *Nrn1*[-/-] cells (*Figure 3—figure supplement 2D*). Moreover, upon restimulation, WT iTreg cells differentiated under NRN1 antibody blockade showed a similar phenotype as *Nrn1*[-/-] cells, with reduced live cell number, reduced Ki67 expression, and increased FOXP3[+] cell proportion among live cells (*Figure 3—figure supplement 2E, F*). These results suggest that *Nrn1* functional deficiency likely contributes to the electrical and metabolic state change observed in *Nrn1*[-/-] iTreg cells.

### *Nrn1* impact effector T cell inflammatory response

CD4[+] T cells can pass through an effector stage on their way to an anergic state (*Huang et al., 2003*). Since *Nrn1* expression is significantly induced after T cell activation (*Figure 1D*), *Nrn1* might influence CD4[+] effector (Te) cell differentiation, affecting anergy development. *Nrn1* may exert different electric changes due to distinct ion channel expression contexts in Te cells than in Tregs. To investigate potential *Nrn1* function in Te cells, we first evaluated *Nrn1*[-/-] Te cell differentiation in vitro. *Nrn1* deficient CD4 Te cells showed increased Ki67 expression, associated with increased cytokine TNFa, Il2, and IFNγ expression upon restimulation (*Figure 4A*). To evaluate *Nrn1*[-/-] Te cell response in vivo, we crossed *Nrn1*[-/-] with FDG mice and generated *Nrn1*[-/-]_FDG and ctrl_FDG mice, which enabled the elimination of endogenous Treg cells (*Figure 4B*). Deleting endogenous FOXP3[+] Treg cells using DT will cause the activation of self-reactive T cells, leading to an autoimmune response (*Kim et al., 2007*; *Nyström et al., 2014*). Upon administration of DT, we observed accelerated weight loss in *Nrn1*[-/-]_FDG mice, reflecting enhanced autoimmune inflammation (*Figure 4C*). Examination of T cell response revealed a significant increase in Ki67 expression and inflammatory cytokine TNFa, IL2, and IFNγ expression among *Nrn1*[-/-] CD4 cells on day 6 post DT treatment (*Figure 4D*), consistent with the findings in vitro. The proportion of FOXP3[+] cells was very low on day 6 post DT treatment and comparable between *Nrn1*[-/-] and the ctrl (*Figure 4E*), suggesting that the differential Te cell response was not due to the impact from Treg cells. Thus, *Nrn1* deficiency enhances Te cell response in vitro and in vivo.

To identify molecular changes responsible for *Nrn1*[-/-] Te phenotype, we compared gene expression between *Nrn1*[-/-] and ctrl Te cells by RNASeq. GSEA and Cytoscape analysis identified a cluster of gene sets on 'membrane repolarization', suggesting that *Nrn1* may also be involved in the regulation of MP under Te context (*Figure 4F*, *Figure 4—source data 1*; *Shannon et al., 2003*; *Subramanian et al., 2005*). While the 'membrane_repolarization' gene set was enriched in *Nrn1*[-/-] (*Figure 4G*), the 'Neurotransmitter receptor activity involved in regulation of postsynaptic membrane potential' gene set was no longer enriched, but the AMPAR subunit *Gria3* expression was still elevated in *Nrn1*[-/-] Te cells (*Figure 4—figure supplement 1A*). Although MP in Te cells was comparable between *Nrn1*[-/-] and ctrl (*Figure 4H*), the 'MF_metal ion transmembrane transporter activity' gene set was significantly

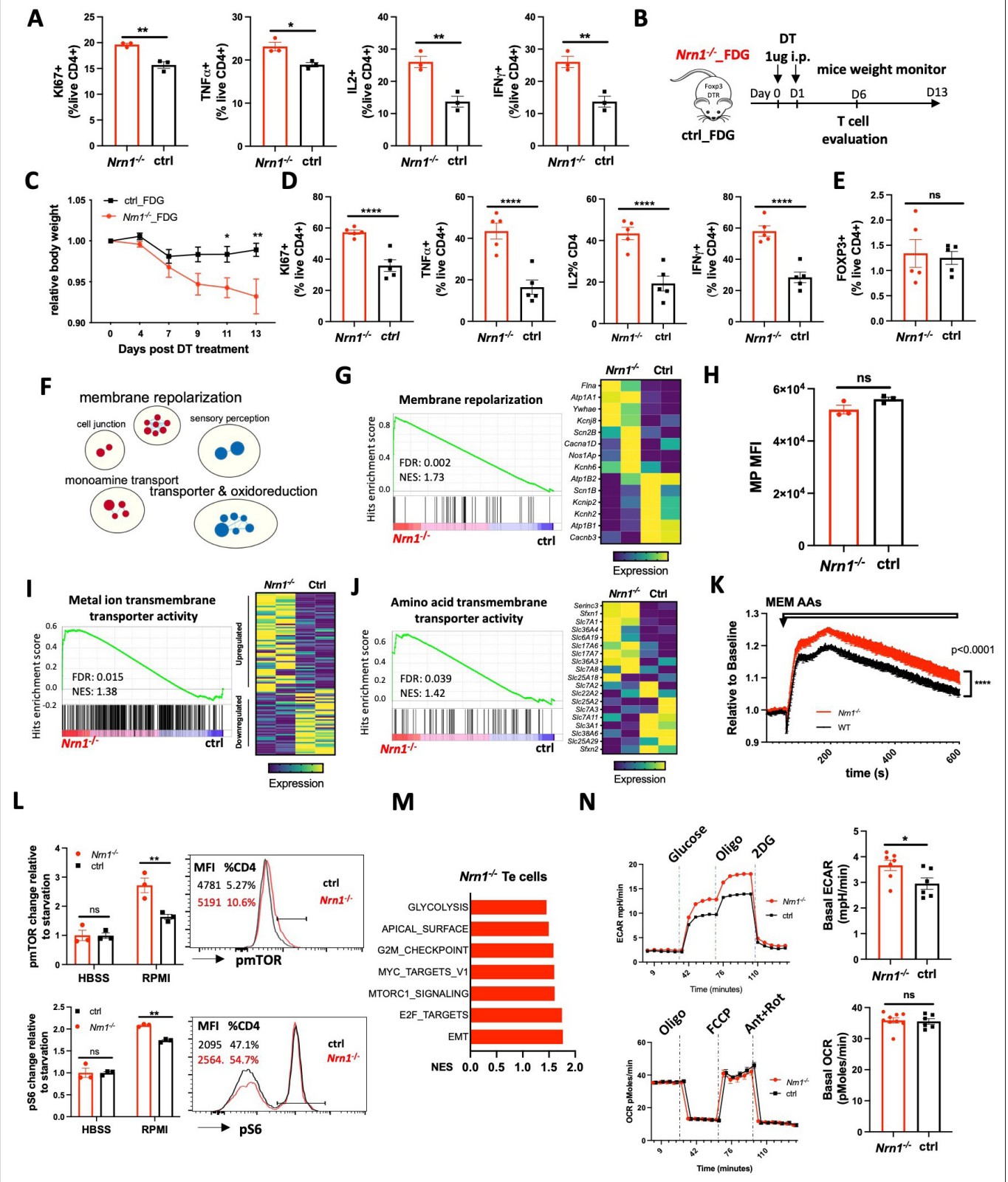

**Figure 4.** *Nrn1* deficiency affects Te cell response. (**A**) Comparison of cell proliferation and cytokine expression in *Nrn1⁻/⁻* and ctrl Te cells. Data represent one of three independent experiments. (**B–E**) An enhanced autoimmune response in *Nrn1⁻/⁻* mice in vivo. (**B**) Experimental scheme. *Nrn1⁻/⁻* mice were crossed with FDG mice and *Nrn1⁻/⁻*_FDG or ctrl_FDG mice were obtained. The autoimmune response was induced by injecting DT i.p. to delete endogenous Treg cells. Mice's weight change was monitored after disease induction. (**C**) Relative body weight change after autoimmune response

*Figure 4 continued on next page*

*Figure 4 continued*

induction. (**D**) Mice were harvested 6 days after DT injection and assessed for ki67, cytokine TNFα, IL2, *and* IFNγ expression in CD4⁺ cells. (**E**) FOXP3 expression among CD4⁺ cells day 6 post DT treatment. n≥5 mice per group. Data represent four independent experiments. (**F–I**) Changes relating to ion balances in Te cells. (**F**) Gene sets clusters from GSEA of GO_MF and GO_Biological process (GO_BP) results in *Nrn1⁻/⁻* and ctrl Te cells (***Figure 4—source data 1***). (**G**) Enrichment of 'GOBP_ membrane repolarization' gene set and enriched gene expression heatmap. (**H**) Membrane potential measurement in Te cells. Data represent two independent experiments. (**I**) Enrichment of 'GOMF_Metal ion transmembrane transporter activity' gene set and heatmap of differential gene expression pattern (***Figure 4—figure supplement 1B***). (**J–N**) Metabolic changes associated with *Nrn1⁻/⁻* Te cell. (**J**) Enrichment of 'GOMF_amino acid transmembrane transporter activity' gene set and differential gene expression heatmap. (**K**) AAs induced MP changes in Te cells. Data represent two independent experiments. (**L**) Measurement of pmTOR and pS6 in Te cells after nutrient sensing. Data represent three independent experiments. (**M**) Enriched Hallmark gene sets (p<0.05, FDR q<0.25). (**N**) Seahorse analysis of extracellular acidification rate (ECAR) and oxygen consumption rate (OCR) in *Nrn1⁻/⁻* and ctrl Te cells. n≥6 technical replicates per group. Data represent three independent experiments. Error bars indicate ± SEM. *p<0.05, **p<0.01, ***p<0.001, ****p<0.0001, unpaired Student's t-test was performed for two-group comparison.

The online version of this article includes the following source data and figure supplement(s) for figure 4:

**Source data 1.** Gene sets enriched in *Nrn1⁻/⁻* and ctrl Te cells.

**Figure supplement 1.** Heatmap of enriched genes in Te cells.

enriched in *Nrn1⁻/⁻* with different gene expression patterns (***Figure 4I***, ***Figure 4—figure supplement 1B***), indicative of different electric state. The significant enrichment of ion channel related genes in *Nrn1⁻/⁻* Te cells was in line with the finding in *Nrn1⁻/⁻* iTreg cells, supporting the notion that *Nrn1* expression may be involved in ion balance and MP modulation.

Examination of nutrient transporters revealed that the 'Amino acid transmembrane transporter activity' gene set was significantly enriched in *Nrn1⁻/⁻* cells than the ctrl (***Figure 4J***). We further examined AA entry-induced cellular MP change in *Nrn1⁻/⁻* and ctrl Te cells. AA entry caused enhanced MP change among *Nrn1⁻/⁻* Te than the ctrl, in contrast with the finding under iTreg cell context (***Figure 4K***). Along with the enrichment of ion channel and nutrient transporter genes (***Figure 4I and J***), we found enhanced mTOR and S6 phosphorylation in *Nrn1⁻/⁻* Te cells (***Figure 4—figure supplement 1C***). We also compared nutrient sensing capability between *Nrn1⁻/⁻* and ctrl Te cells, as outlined in ***Figure 3I***. *Nrn1⁻/⁻* Te showed increased mTOR and S6 phosphorylation after sensing ions and nutrients in RPMI medium (***Figure 4L***), confirming the differential impact of ions and nutrients on *Nrn1⁻/⁻* and ctrl Te cells. GSEA on Hallmark collection showed enrichment of mTORC1 signaling gene set (***Figure 4M***), corroborating with increased pmTOR and pS6 detection in *Nrn1⁻/⁻* Te cells. Along with increased mTORC1 signaling, *Nrn1⁻/⁻* Te cells also showed enrichment of gene sets on glycolysis and proliferation (***Figure 4M***). Evaluation of metabolic changes by seahorse confirmed increased glycolysis in *Nrn1⁻/⁻* cells, while the OCR remained comparable between *Nrn1⁻/⁻* and ctrl (***Figure 4N***). These in vitro studies on Te cells indicate that Nrn1 deficiency resulted in the dysregulation of the electrolyte and nutrient transport program, impacting Te cell nutrient sensing, metabolic state, and the outcome of inflammatory response.

### *Nrn1* deficiency exacerbates autoimmune disease

The coordinated reaction of Treg and Te cells contributes to the outcome of the immune response. We employed the experimental autoimmune encephalomyelitis (EAE), the murine model of multiple sclerosis (MS), to evaluate the overall impact of *Nrn1* on autoimmune disease development. Upon EAE induction, the incidence and time to EAE onset in Nrn1⁻/⁻ mice were comparable to the ctrl mice, but the severity, disease persistence, and body weight loss were increased in *Nrn1⁻/⁻* mice (***Figure 5A***). Exacerbated EAE was associated with significantly increased CD45⁺ cell infiltrates, increased CD4⁺ cell number, increased proportion of MOG-specific CD4 cells, and reduced proportion of FOXP3⁺ CD4 cells in the *Nrn1⁻/⁻* spinal cord (***Figure 5B–E***). Moreover, we also observed increased proportions of IFNγ⁺ and IL17⁺ CD4 cells in *Nrn1⁻/⁻* mice remaining in the draining lymph node compared to the ctrl mice (***Figure 5F***). Thus, the results from EAE corroborated with earlier data and confirmed the important role of *Nrn1* in establishing immune tolerance and modulating autoimmunity.

## Discussion

T cell expansion and functional development depend on adaptive electric and metabolic changes, maintaining electrolyte balances, and appropriate nutrient uptake. The negative charge of the plasma membrane, ion channel expression pattern, and function are key characteristics associated with

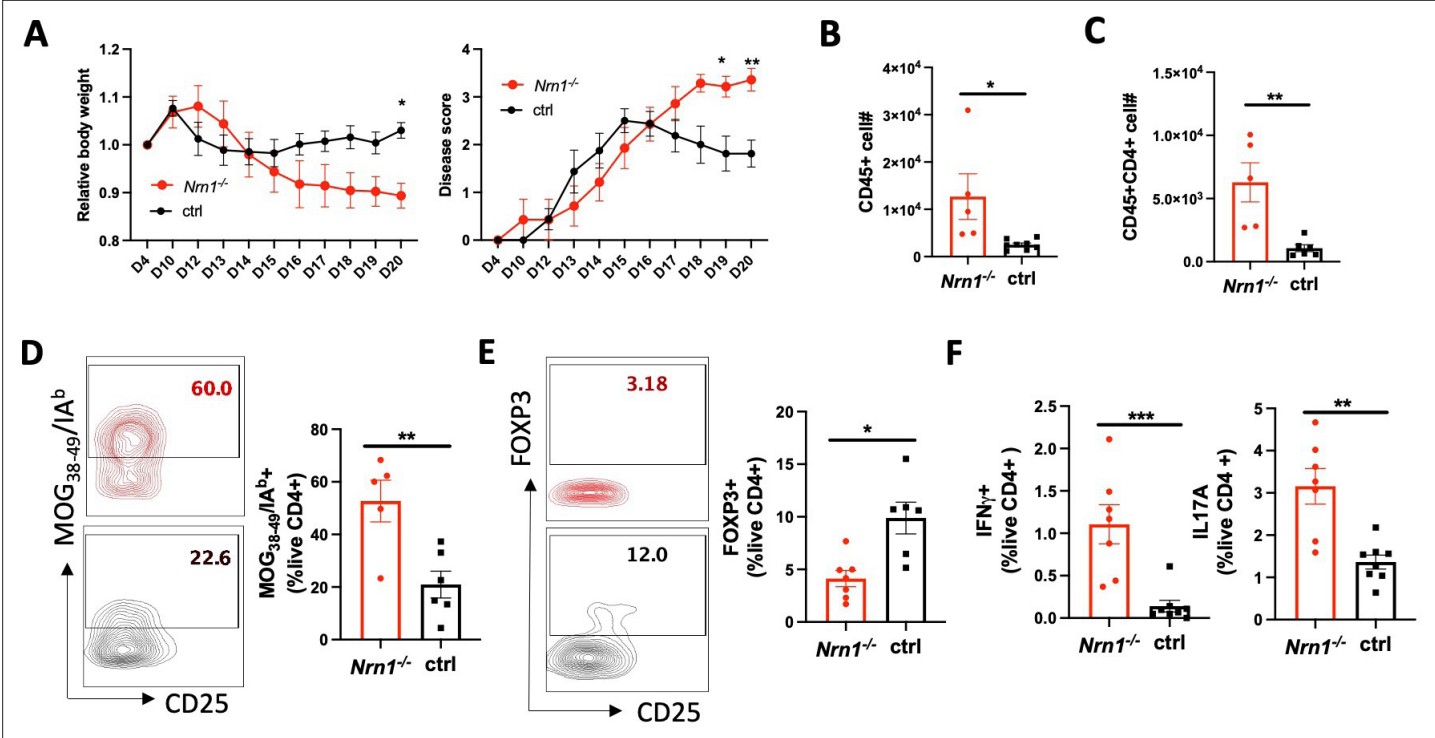

**Figure 5.** *Nrn1* deficiency exacerbates autoimmune EAE disease. (**A**) Aggravated body weight loss and protracted EAE disease in *Nrn1*⁻/⁻ mice. (**B**) CD45⁺ cell number in the spinal cord infiltrates. (**C**) CD4⁺ cell number in the spinal cord infiltrates. (**D**) Mog₃₈₋₄₉/IA^b tetramer staining of spinal cord infiltrating CD4 cells. (**E**) FOXP3⁺ proportion among CD4⁺ cells in spinal cord infiltrates. (**F**) IFNγ+and IL17⁺ cell proportion among CD4⁺ cells in draining lymph nodes. n≥5 mice per group. Data represent three independent experiments. The p value was calculated by 2way ANOVA for (**A**). The p-value was calculated by the unpaired student t-test for (**B–F**). *p<0.05, **p<0.01.

the cellular electric state in different systems, impacting cell proliferation and function (*Blackiston et al., 2009*; *Emmons-Bell and Hariharan, 2021*; *Kiefer et al., 1980*; *Monroe and Cambier, 1983*; *Sundelacruz et al., 2009*). The electrolytes and nutrients, including amino acids, metabolites, and small peptides transported through ion channels and nutrient transporters, are also regulators and signaling agents impacting the choice of cellular metabolic pathways and functional outcomes (*Hamill et al., 2020*). In this study, we report that the neurotropic factor *Nrn1* expression influences CD4 T cell MP, ion channels, and nutrient transporter expression patterns, contributing to differential metabolic states in Treg and Te cells. *Nrn1* deficiency compromises Treg cell expansion and suppression while enhancing Te cell inflammatory response, exacerbating autoimmune disease.

Bioelectric controls have been defined as a type of epigenetics that can control information residing outside of genomic sequence (*Levin, 2021*). The sum of ion channels and pump activity generates the ionic gradient across the cell membrane, establishing the MP level and bioelectric state. Cells with the same MP can have different ion compositions, and the same ion channel may have a differential impact on MP when in combination with different ion channels (*Abdul Kadir et al., 2018*). Consistent with this notion, *Nrn1* deficiency has differential impacts on the cellular electric state under the Treg and Te cells with different ion channel combinations. Altered MP was detected in *Nrn1* deficient Treg cells (*Figure 3E*), while comparable MP was observed between *Nrn1*⁻/⁻ and ctrl Te cells (*Figure 4H*). The MP level determined by ion channels and pump activity can influence the nutrient transport pattern, establishing a metabolic and functional state matching the MP level (*Blackiston et al., 2009*; *Emmons-Bell and Hariharan, 2021*; *Kiefer et al., 1980*; *Monroe and Cambier, 1983*; *Sundelacruz et al., 2009*; *Yu et al., 2022*). Yu et al. reported that macrophage MP modulates plasma membrane phospholipid dynamics and facilitates cell surface retention of nutrient transporters, thus supporting nutrient uptake and impacting the inflammatory response (*Yu et al., 2022*). Nutrient transport is key to T cell fate decisions and has been considered signal 4 to T cell fate choices (*Chapman and Chi, 2022*; *Long et al., 2021*). The changes in ion channel related gene expression and MP level

in *Nrn1*[-/-] cells were accompanied by differential expression of AA transporter genes and nutrient sensing activity that impacted mTORC1 pathway activation and cellular glycolytic state (*Figures 3 and 4*). These results corroborate previous observations on the connection of MP in nutrient acquisition and metabolic change and support the role of *Nrn1* in coordinating T cell electric and metabolic adaptation (*Yu et al., 2022*).

Although *Nrn1*, as a small GPI-anchored protein, does not have channel activity by itself, it has been identified as one of the components in the AMPAR complex (*Pandya et al., 2018*; *Schwenk et al., 2012*; *Subramanian et al., 2019*). Na[+]-influx through the AMPA type ionotropic glutamate receptor can quickly depolarize the postsynaptic compartment and potentiate synaptic transmission in neurons. We have observed increased expression of AMPAR subunits in *Nrn1*[-/-] iTreg and Te cells (*Figure 3D*, *Figure 4—figure supplement 1*), implicating potential change in AMPAR activity in *Nrn1*[-/-] under Treg and Te cell context. Glutamate secreted by proliferating cells may influence T cell function through AMPAR. High glutamate levels are detected at the autoimmune disease site and tumor interstitial fluid (*Bonnet et al., 2020*; *McNearney et al., 2004*; *Sullivan et al., 2019*). Moreover, AMPAR has been implicated in exacerbating autoimmune disease (*Bonnet et al., 2015*; *Sarchielli et al., 2007*). The increased expression of AMPAR subunits in *Nrn1*[-/-] cells supports the potential connection of *Nrn1* and AMPAR and warrants future investigation on the possibility that *Nrn1* functions through AMPAR, impacting T cell electric change. Besides AMPAR, *Nrn1* has been reported to function through the insulin receptor and fibroblast growth factor pathway (*Shimada et al., 2016*; *Yao et al., 2012*). Subramanian et al have suggested that rather than a traditional ligand with its cognate receptor, *Nrn1* may function as an adaptor to receptors to perform diverse cell-type-specific functions (*Subramanian et al., 2019*). Our results do not rule out these possibilities.

Overall, we found that *Nrn1* expression in Treg and Te cells can impact cellular electric state, nutrient sensing, and metabolism in a cell context-dependent manner. The predominant enrichment of ion channel related gene sets in both Treg and Te cell context underscores the importance of *Nrn1* in modulating ion balance and MP. The changes in ion channels and nutrient transporter expression in Treg and Te cells and associated functional consequences highlight the importance of *Nrn1* in coordinating cell metabolic changes through channels and transporters during the adaptive response and contribute to the balance of tolerance and immunity.

# Materials and methods

**Key resources table**

| Reagent type (species) or resource | Designation | Source or reference | Identifiers | Additional information |
|---|---|---|---|---|
| Antibody | Purified anti-mouse CD3 | Biolegend | Cat. No. 100202 | 5 ug/ml for stimulation |
| Antibody | APC anti-mouse CD4 | Biolegend | Cat. No. 100516 | FACS (1:500) |
| Antibody | FITC anti-mouse CD4 | Biolegend | Cat. No. 100706 | FACS (1:500) |
| Antibody | PE/Cyanine7 anti-mouse CD25 | Biolegend | Cat. No. 102016 | FACS (1:500) |
| Antibody | Pacific Blue anti-mouse CD45.1 | Biolegend | Cat. No. 110722 | FACS (1:500) |
| Antibody | APC anti-mouse CD45.2 Antibody | Biolegend | Cat. No. 109814 | FACS (1:500) |
| Antibody | APC/Cyanine7 anti-mouse CD62L | Biolegend | Cat. No. 104428 | FACS (1:500) |
| Antibody | PE anti-mouse CD73 Antibody | Biolegend | Cat. No. 127206 | FACS (1:400) |
| Antibody | PerCP/Cyanine5.5 anti-mouse CD90.1 (Thy1.1) | Biolegend | Cat. No. 109004 | FACS (1:500) |
| Antibody | APC/Cyanine7 anti-mouse CD90.2 (Thy1.2) | Biolegend | Cat. No. 105328 | FACS (1:500) |

*Continued on next page*

*Continued*

| Reagent type (species) or resource | Designation | Source or reference | Identifiers | Additional information |
|---|---|---|---|---|
| Antibody | PE anti-mouse TCR Vβ5.1, 5.2 | Biolegend | Cat. No. 139504 | FACS (1:500) |
| Antibody | APC/Cyanine7 anti-mouse CD279 (PD-1) | Biolegend | Cat. No. 135224 | FACS (1:500) |
| Antibody | Alexa Fluor 700 anti-mouse IFN-g | Biolegend | Cat. No. 505824 | FACS (1:500) |
| Antibody | PE anti-mouse IL-17A | Biolegend | Cat. No. 506904 | FACS (1:500) |
| Antibody | Alexa Fluor 700 anti-mouse TNF-α | Biolegend | Cat. No. 506338 | FACS (1:500) |
| Antibody | Alexa Fluor 594 anti-T-bet | Biolegend | Cat. No. 644833 | FACS (1:300) |
| Antibody | PerCP/Cyanine5.5 anti-mouse Ki-67 A | Biolegend | Cat. No. 652424 | FACS (1:500) |
| Antibody | PE Rat Anti-Mouse CD44 | BD Bioscience | Cat. No. 561860 | FACS (1:500) |
| Antibody | BV605 Rat Anti-Mouse CD45 | BD Bioscience | Cat; No. 563053 | FACS (1:500) |
| Antibody | PE Hamster Anti-Mouse CD69 | BD Bioscience | Cat. No: 553237 | FACS (1:500) |
| Antibody | PE FOXP3 Monoclonal Antibody (FJK-16s) | ThermoFisher eBioscience | Cat. No. 12-5773-82 | FACS (1:300) |
| Antibody | Biotin anti-NRN1 (1 A10) | custom made | A&G Pharmaceutical | FACS (1:200) |
| Antibody | anti-NRN1 (1D6) | custom made | A&G Pharmaceutical | 10 ug/ml for blocking |
| Antibody | purified antiCD28 | Bio-X Cell | Cat. No.BE0015-1 | 2 ug/ml ofr stimulation |
| Antibody | purified anti-mouse IL-4 | Bio-X Cell | Cat. No.BE0045 | 5 ug/ml for blocking |
| Antibody | purified anti-mouse IFNg | Bio-X Cell | Ca. No.BE0055 | 5 ug/ml for blocking |
| Chemical compound, drug | Fluo-4, AM, | Invitrogen | Cat. No.F14201 | 2 ug/ml |
| Chemical compound, drug | Thapsigargin | Invitrogen | Cat.No.T7458 | 1 uM |
| Chemical compound, drug | Oligomycin | Sigma | Cat. No.O4876-5MG | 1 uM |
| Chemical compound, drug | 2-Deoxy-D-glucose | Sigma | Cat. No.D8375-1G | 50 mM |
| Chemical compound, drug | FCCP | Sigma | Cat. No.SML2959 | 2 uM |
| Chemical compound, drug | Rotenone | Sigma | Cat. No.557368–1 GM | 1 uM |
| Chemical compound, drug | Antimycin A | Sigma | Cat. No.A8674-25MG | 1 uM |
| Chemical compound, drug | BD Difco Adjuvants | Fisher | Cat. No.DF3114-33-8 | 500 ug/mouse |
| Peptide, recombinant protein | Human IL-2 Recombinant Protein | peproTech | Cat No.200-02-50UG | 100 ng/ml |
| Peptide, recombinant protein | Human TGF-beta 1 Recombinant | peproTech | Cat No. 100-21-10UG | 10 ng/ml |

*Continued on next page*

*Continued*

| Reagent type (species) or resource | Designation | Source or reference | Identifiers | Additional information |
|---|---|---|---|---|
| Peptide, recombinant protein | Pertussis Toxin from B. pertussis, | List Laboratory | Cat. No.180 | 400 ng/mouse |
| Peptide, recombinant protein | OVA$_{323-339}$ | GeneScript | Cat. No.RP10610 | 100 ug/mouse |
| Peptide, recombinant protein | MOG$_{35-55}$ | GeneScript | Cat. No.RP10245 | 200 ug/mouse |
| Strain, strain background | *Nrn1$^{-/-}$* mice backcrossed to C57/BL6 background | The Jackson Laboratory | RRID:IMSR_JAX:018402 | |
| Strain, strain background | FOXP3DTRGFP, C57/BL6 background | The Jackson Laboratory | RRID:IMSR_JAX:016958 | |
| Strain, strain background | TCRa$^{-/-}$, C57/BL6 | The Jackson Laboratory | RRID:IMSR_JAX:002116 | |
| Strain, strain background | OTII, C57/BL6 | Jonathan Powell, parental strain: The Jackson Laboratory | RRID:IMSR_JAX:004194 | |
| Strain, strain background | Rag2-/-, C57/BL6 | Pardoll Lab, parental strain:The Jackson Laboratory | RRID:IMSR_JAX:008449 | |
| Strain, strain background | 6.5 TCR transgenic mice, B10.D2 | Pardoll Lab, parental strain:von Boehmer Lab | | |
| Strain, strain background | C3HA transgenic mice, B10.D2 | Pardoll Lab | | |
| *Sequence-based reagent* | *Nrn1 Forward* | IDT | GCGGTGCAAATAGCTTACCTG | |
| Sequence-based reagent | *Nrn1 Reverse* | IDT | CGGTCTTGATGTTCGTCTTGTC | |
| Software, Algorithims | STAR aligner | *Dobin et al., 2013* | https://www.ncbi.nlm.nih.gov/pubmed/23104886 | |
| Software, Algorithims | HTSeq | *Anders et al., 2015* | https://pypi.org/project/HTSeq/ | |
| Software, Algorithims | DESeq2 | *Love et al., 2014* | https://bioconductor.org/packages/devel/bioc/vignettes/DESeq2/inst/doc/DESeq2.html | |
| Software, Algorithims | GSEA | *Subramanian et al., 2005* | https://www.gsea-msigdb.org/gsea/index.jsp | |
| Software, Algorithims | Cytoscape | *Shannon et al., 2003* | https://cytoscape.org/ | |
| Software, Algorithims | FlowJo 10.5.3 | BD Bioscience | https://www.flowjo.com/solutions/flowjo | |
| Software, Algorithims | Prism 10 | GraphPad | https://www.graphpad.com/ | |

## Mouse models

*The Nrn1$^{-/-}$ mice* (**Fujino et al., 2011**), FOXP3DTRGFP *(FDG)* (**Kim et al., 2007**), *and TCRα$^{-/-}$* mice were obtained from the Jackson Laboratory. OTII mice on Thy1.1$^+$ background were kindly provided by Dr. Jonathan Powell. *Rag2$^{-/-}$* mice were maintained in our mouse facility. 6.5 TCR transgenic mice specific for HA antigen and C3HA mice (both on the B10.D2 background) have been described previously (**Huang et al., 2004**). *Nrn1$^{-/-}$* mice were crossed with OTII mice to generate *Nrn1$^{-/-}$_OTII$^+$* mice, ctrl_OTII$^+$ mice. *Nrn1$^{-/-}$* mice were also crossed with FDG mice to generate *Nrn1$^{-/-}$_FDG* and ctrl_FDG mice. All mice colonies were maintained in accordance with the guidelines of Johns Hopkins University and the institutional animal care and use committee.

## Antibodies and reagents

We have used the following antibodies: Anti-CD3 (17A2), anti-CD4 (RM4-5), anti-CD8a (53–6.7), anti-CD25 (PC61), anti-CD45.1 (A20), anti-CD45.2 (104), anti-CD62L (MEL-14), CD73 (TY/111.8), anti-CD90.1 (OX-7), anti-CD90.2 (30-H12), anti-TCR Vβ5.1, 5.2 (MR9-4), anti-PD1 (29 F.1A12), anti-IFNγ (XMG1.2), anti-IL17a (TC11-18H10.1), anti-TNFa (MP6-XT22), anti-Tbet (4B10), anti-Ki67 (16A8) were purchased from Biolegend. Anti-CD44 (IM7), CD45 (30-F11), anti-CD69(H1.2F3) were purchased from BD Bioscience. Anti-FOXP3 (FJK-16s) was purchased from eBioscience. The flow cytometry data were collected using BD Celesta (BD Biosciences) or Attune Flow Cytometers (Thermo Fisher). Data were analyzed using FlowJo (Tree Star) software.

Mouse monoclonal anti-NRN1 antibody (Ab) against NRN1 was custom-made (A&G Pharmaceutical). The specificity of anti-NRN1 Ab was confirmed by ELISA, cell surface staining of NRN1 transfected 293T cells, and western blot of NRN1 recombinant protein and brain protein lysate from WT mice or $Nrn1^{-/-}$ mice (data not shown). $OVA_{323-339}$ peptide and $MOG_{35-55}$ was purchased from GeneScript. Incomplete Freund's adjuvant (IFA) and *Mycobacterium tuberculosis* H37Ra (killed and desiccated) were purchased from Difco. Pertussis toxin was purchased from List Biological Laboratories and diphtheria toxin was obtained from Millipore-Sigma.

## Cell purification and culture

Naive CD4 cells were isolated from the spleen and peripheral lymph node by a magnetic bead-based purification according to the manufacturer's instruction (Miltenyi Biotech). Purified CD4 cells were stimulated with plate-bound anti-CD3 (5 µg/ml, Bio-X-Cell) and anti-CD28 (2 µg/ml, Bio-X-Cell) for 3 days, in RPMI1640 medium supplemented with 10%FBS, HEPES, penicillin/streptomycin, MEM Non-Essential Amino Acids, and β-mercaptoethanol. For iTreg cell differentiation, cells were stimulated in the presence of human IL2 (100 u/ml, PeproTech), human TGFβ (10 ng/ml, PeproTech), anti-IL4, and anti-IFNγ antibody (5 µg/ml, Clone 11B11 and clone XMG1.2, Bio-X-Cell) in 10% RPMI medium. $CD4^+$ Te cells were differentiated without additional cytokine or antibody for 3 days, followed by additional culture for 2 days in IL2 100 u/ml in 10%RPMI medium. nTreg cells were isolated by sorting from the FDG $CD4^+$ fraction based on $FOXP3^+GFP$ and CD25 expression ($CD4^+CD25^+GFP^+$). Alternatively, nTreg cells were enriched from CD4 cells by positive selection using the $CD4^+CD25^+$ Regulatory T Cell Isolation Kit from Miltenyi.

## Self-antigen induced tolerance model

$1x10^6$ HA-specific $Thy1.1^+$ 6.5 CD4 cells from donor mice on a B10.D2 background were transferred into C3-HA recipient mice, where HA is expressed as self-antigen in the lung; or into WT B10.D2 mice followed by infection with Vac-HA virus ($1x10^6$ pfu). HA-reactive T cells were recovered from the lung-draining lymph node of C3-HA host mice or WT B10.D2 Vac-HA infected mice at indicated time points by cell sorting. RNA from sorted cells was used for qRT-PCR assay examining *Nrn1* expression.

## Peptide-induced T cell anergy model

$5x10^5$ Polyclonal Treg cells from $CD45.1^+$ C57BL/6 mice were mixed with $5x10^6$ $thy1.1^+$ OTII cells from $Nrn1^{-/-}$_OTII or ctrl_OTII mice and transferred by *i.v.* injection into $TCRα^{-/-}$ mice. 100 µg of $OVA_{323–339}$ dissolved in PBS was administered *i.v.* on days 1, 4, and 7 after cell transfer. Host mice were harvested on day 13 after cell transfer, and cells from the lymph node and spleen were further analyzed.

## In vivo Treg suppression assay

nTreg cells from $CD45.2^+CD45.1^-$ $Nrn1^{-/-}$ or ctrl mice ($5x10^5$/mouse) in conjunction with $CD45.1^+$ splenocytes ($2x10^6$/mouse) from FDG mice were cotransferred i.p. into $Rag2^{-/-}$ mice. The $CD45.1^+$ splenocytes were obtained from FDG mice pretreated with DT for 2 days to deplete Treg cells. Treg suppression toward $CD45.1^+$ responder cells was assessed on day 7 post cell transfer. Alternatively, 7 days after cell transfer, $Rag2^{-/-}$ hosts were challenged with an *i.d.* inoculation of B16F10 cells ($1x10^5$). Tumor growth was monitored daily. Treg-mediated suppression toward anti-tumor response was assessed by harvesting mice day 18–21 post-tumor inoculation.

## Induction of autoimmunity by transient Treg depletion

To induce autoimmunity in $Nrn1^{-/-}$_FDG and ctrl_FDG mice, 1 µg/mouse of DT was administered i.p. for 2 consecutive days, and the weight loss of treated mice was observed over time.

## EAE induction

EAE was induced in mice by subcutaneous injection of 200 µg MOG$_{35-55}$ peptide with 500 µg *M. tuberculosis* strain H37Ra (Difco) emulsified in incomplete Freund Adjuvant oil in 200 µl volume into the flanks at two different sites. In addition, the mice received 400 ng pertussis toxin (PTX; List Biological Laboratories) *i.p.* at the time of immunization and 48 hr later. Clinical signs of EAE were assessed daily according to the standard 5-point scale (*Miller et al., 2007*): normal mouse; 0, no overt signs of disease; 1, limp tail; 2, limp tail plus hindlimb weakness; 3, total hindlimb paralysis; 4, hindlimb paralysis plus 75% of body paralysis (forelimb paralysis/weakness); 5, moribund.

## ELISA

MaxiSorp ELISA plates (Thermo Fisher Scientific Nunc) were coated with 100 µl of 1 µg/ml anti-mIL-2 (BD Pharmingen #554424) at 4 °C overnight. Coated plates were blocked with 200 µl of blocking solution (10%FBS in PBS) for 1 hr at room temperature (RT) followed by incubation of culture supernatant and mIL-2 at different concentrations as standard. After 1 hr, plates were washed and incubated with anti-mIL-2-biotin (BD Pharmingen #554426) at RT for 1 hr. After 1 hr, plates were incubated with 100 µl of horseradish peroxidase-labeled avidin (Vector Laboratory, #A-2004) 1 µg/ml for 30 min. After washing, samples were developed using the KPL TMB Peroxidase substrate system (Seracare #5120–0047) and read at 405 or 450 nm after the addition of the stop solution.

## Quantitative RT-PCR

RNA was isolated using the RNeasy Micro Kit (QIAGEN 70004) following the manufacturer's instructions. RNA was converted to cDNA using the High-Capacity cDNA Reverse Transcription Kit (Thermo Fisher Scientific #4368814) according to the manufacturer's instructions. The primers of murine genes were purchased from Integrated DNA Technology (IDT). qPCR was performed using the PowerUp SYBR Green Master Mix (Thermo Fisher Scientific #A25780) and the Applied Biosystems StepOnePlus 96-well real-time PCR system. Gene expression levels were calculated based on the Delta-Delta Ct relative quantification method. Primers used for *Nrn1* PCR were as follows: GCGGTGCAAATAGCTT ACCTG (forward); CGGTCTTGATGTTCGTCTTGTC (reverse).

## Ca$^{++}$ flux and Membrane potential measurement

To measure Ca$^{++}$ flux, CD4 cells were loaded with Fluo4 dye at 2 µM in the complete cell culture medium at 37 °C for 30 min. Cells were washed and resuspended in HBSS Ca$^{++}$-free medium and plated into 384 well glass bottom assay plate (minimum of 4 wells per sample). Ca$^{++}$ flux was measured using the FDSS6000 system (Hamamatsu Photonics). To measure store-operated calcium entry (SOCE), after the recording of the baseline T cells Ca$^{++}$ fluorescent for 1 min, thapsigargin (TG) was added to induce store Ca$^{++}$ depletion, followed by the addition of Ca$^{++}$ 2µM in the extracellular medium to observe Ca$^{++}$ cellular entry.

Membrane potential was measured using FLIPR Membrane Potential Assay kit (Molecular devices) according to the manufacturer's instructions. Specifically, T cells were loaded with FLIPR dye by adding an equal volume of FLIPR dye to the cells and incubated at 37 °C for 30 min. Relative membrane potential was measured by detecting FLIPR dye incorporation using flow cytometry.

To measure changes of MP after AAs transport, T cells were plated and loaded with FLIPR dye at 37 °C for 30 min in 384-well glass bottom assay plate (minimum of 6 wells per sample). After recording the baseline T cell MP for 1 min, MEM AAs (Gibco MEM Amino Acids #11130–051) were injected into each well, and the change of MP was recorded for 5 min.

## Extracellular flux analysis (Seahorse assays)

Real-time measurements of extracellular acidification rate (ECAR) and oxygen consumption rate (OCR) were performed using an XFe-96 Bioanalyser (Agilent). T cells (2×10$^5$ cells per well; minimum of four wells per sample) were spun into previously poly-d-lysine-coated 96-well plates (Seahorse) in complete RPMI-1640 medium. ECAR was measured in RPMI medium in basal condition and

in response to 25 mM glucose, 1 µM oligomycin, and 50 mM of 2-DG (all from Sigma-Aldrich). OCR was measured in RPMI medium supplemented with 25 mM glucose, 2 mM L-glutamine, and 1 mM sodium pyruvate, under basal condition and in response to 1 µM oligomycin, 1.5 µM of carbonylcyanide-4-(trifluoromethoxy)-phenylhydrazone (FCCP) and 1 µM of rotenone and antimycin (all from Sigma-Aldrich).

## RNAseq and data analysis

RNASeq samples: 1. Anergic T cell analysis. Ctrl and *Nrn1*[-/-] OTII cells were sorted from the host mice (n=3 per group). 2. iTreg cell analysis. In vitro differentiated *Nrn1*[-/-] and ctrl iTreg cells were replated in resting condition (IL2 100 u/ml) or stimulation condition (IL2 100 u/ml and aCD3 5 µg/ml). Cells were harvested 20 hr after replating for RNASeq analysis. 3. Effector T cells. *Nrn1*[-/-] and ctrl CD4 Tn cells were activated for 3 days (aCD3 5 µg/ml, aCD28 2 µg/ml), followed by replating in IL2 medium (100 u/ml). Te cells were harvested two days after replating and subjected to RNASeq analysis.

RNA-sequencing analysis was performed by Admera Health (South Plainfield, NJ). Read quality was assessed with FastQC and aligned to the *Mus musculus* genome (Ensembl GRCm38) using STAR aligner (version 2.6.0; *Dobin et al., 2013*). Aligned reads were counted using HTSeq (version 0.9.0; *Anders et al., 2015*), and the counts were loaded into R (The R Foundation). DESeq2 package (version 1.24.0; *Love et al., 2014*) was used to normalize the raw counts. GSEA was performed using public gene sets (HALLMARK, and GO; *Subramanian et al., 2005*). Cytoscape was used to display enriched gene sets cluster (*Shannon et al., 2003*).

Statistical analysis. All numerical data were processed using Graph Pad Prism 10. Data are expressed as the mean +/-the SEM, or as stated. Statistical comparisons were made using an unpaired student t-test or ANOVA with multiple comparison tests where 0.05 was considered significant, and a normal distribution was assumed. The p values are represented as follows: * $p<0.05$; ** $p<0.01$; *** $p<0.001$, **** $p<0.0001$.

## Acknowledgements

This research is supported by grants from the Bloomberg-Kimmel Institute of JHU, the Melanoma Research Alliance, the National Institutes of Health (RO1AI099300 and RO1AI089830), and the Department of Defense (PC130767). JB's research was supported by a Crohn's and Colitis Foundation of America Research Fellowship, the Melanoma Research Foundation, and NCI grant P30CA016056. We thank Dennis Gong for data processing and critical reading of the manuscript. We thank Dr. Elly Nedivi for providing polyclonal *Nrn1* antibody and Dr. Fan Pan for reagent support. We thank Drs. Franck Housseau, Chien-Fu Hung for the constructive discussion of the project and the manuscript. We thank Drs. Hao Shi and Hongbo Chi for critical reading of the manuscript and helpful suggestions. We thank Dr. Rachel Helm for manuscript editing. We thank Drs. Richard L Huganir, Bian Liu and Hana Goldschmidt for constructive discussion on Nrn1 and AMPAR connection.

## Additional information

### Funding

| Funder | Grant reference number | Author |
|---|---|---|
| National Institute of General Medical Sciences | RO1AI099300 | Drew Pardoll |
| National Institute of General Medical Sciences | RO0AI089830 | Drew Pardoll |
| Department of Defense | PC130767 | Drew Pardoll |
| NCI | P30CA016056 | Joseph Barbi |

The funders had no role in study design, data collection and interpretation, or the decision to submit the work for publication.

## Author contributions
Hong Yu, Conceptualization, Data curation, Formal analysis, Supervision, Validation, Investigation, Methodology, Writing – original draft, Project administration, Writing – review and editing; Hiroshi Nishio, Joseph Barbi, Data curation, Formal analysis, Investigation, Methodology; Marisa Mitchell-Flack, Paolo DA Vignali, Andriana Lebid, Kwang-Yu Chang, Data curation, Formal analysis, Investigation; Ying Zheng, Data curation, Formal analysis; Juan Fu, Data curation, Investigation, Methodology; Makenzie Higgins, Lee Blosser, Ada Tam, Charles Drake, Methodology; Ching-Tai Huang, Data curation, Methodology; Xuehong Zhang, Zhiguang Li, Data curation; Drew Pardoll, Resources, Supervision, Funding acquisition, Project administration, Writing – review and editing

## Author ORCIDs
Hong Yu https://orcid.org/0000-0002-8778-9870

## Ethics
This study was performed in strict accordance with the recommendation in the Care and Use of Laboratory Animals of the National Institute of Health. All of the animals were handled according to approved institutional animal care and use committee (IACUC) protocol (M019M233) of Johns Hopkins University.

Reviewer #1 (Public review): https://doi.org/10.7554/eLife.96812.3.sa1
Reviewer #2 (Public review): https://doi.org/10.7554/eLife.96812.3.sa2
Author response https://doi.org/10.7554/eLife.96812.3.sa3

# Additional files

## Supplementary files
MDAR checklist

## Data availability
RNA sequencing data has been deposited under the GEO accession numbers GSE121908 and GSE224083.

The following datasets were generated:

| Author(s) | Year | Dataset title | Dataset URL | Database and Identifier |
|---|---|---|---|---|
| Pardoll D, Nishio H, Yu H | 2019 | RNA-sequencing on Nrn1-/- and Nrn1+/- OTII cells recovered from OVA-peptide induced anergy TCRa-/- host mice | https://www.ncbi.nlm.nih.gov/geo/query/acc.cgi?acc=GSE121908 | NCBI Gene Expression Omnibus, GSE121908 |
| Yu H | 2024 | The neurotrophic factor neuritin impacts T cell electrical and metabolic state for the balance of tolerance and immunity | https://www.ncbi.nlm.nih.gov/geo/query/acc.cgi?acc=GSE224083 | NCBI Gene Expression Omnibus, GSE224083 |

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
