## [Editor Report · eLife Assessment]

The neurotrophic factor Neuritin can moderate T-cell tolerance and immunity through both regulatory T (Treg) and effector T cells, promoting Treg cell expansion and suppression while dampening effector T cells to mediate the inflammatory response. Neuritin expression influences the membrane potential, ion channels, and nutrient transporter expression patterns of CD4+ T cells, contributing to differential metabolic states in Treg and effector T cells. These findings are **solid** and **important** for understanding immune regulation involving Treg cells and effector T cells.

---

## [Referee Report · Reviewer #1 (Public review)]

The manuscript by Yu et al seeks to investigate the role of neuritin (Nrn1), identified as a marker of anergic cells, in the biology of regulatory (Tregs) and conventional (Tconv) T cells. Although the role of Nrn1 expressed by Tregs has already been explored (Gonzalez-Figueroa 2021 cited in the manuscript), this manuscript shows original new data suggesting that this molecule would be important in promoting Treg function and inhibiting Tconv effector function by acting at the level of membrane potential and molecule transport across the plasma membrane. However, multiple models have been used, but none has been studied thoroughly enough to provide really conclusive and unambiguous data. For example, 5 different models were used to study T cells in vivo. It would have been preferable to use fewer, but to go further in the study of mechanisms. In the absence of more in-depth study, the conclusions drawn by the authors are often open to questions. Major points concern the fact that there are not enough biological replicates for most experiments and some critical controls and data are lacking. Also, the authors have used iTregs rather than nTregs for many experiments (see below). This is unfortunate because the role of neuritin in T cell biology studied here is new and interesting.

Major points (in the order in which they appear in the text).

(1) A real weakness of this work is the fact that in most of the results shown, there are few biological replicates with differences that are often small between Ctrl and Nrn1 -/-. The systematic use of student's t test may lead to think that the differences are significant, which is often misleading given the small number of samples, which makes it impossible to know whether the distributions are Gaussian and whether a parametric test can be used. RNAseq bulk data are based on biological duplicates, which is open to criticism.

(2) The authors use Nrn1+/+ and Nrn1+/- cells indiscriminately as control cells on the basis of similar biology between Nrn1+/+ and Nrn1+/- cells at homeostasis. However, it is quite possible that the Nrn1+/- cells have a phenotype in situations of in vitro activation or in vivo inflammation (cancer, EAE). It would be important to discriminate Nrn1+/- and Nrn1+/+ cells in the data or to show that both cell types have the same phenotype in these conditions too.

(3) Fig 1A-D. Since the authors are using the Nrp1 KO mice, it would be important to confirm the specificity of the anti-Nrn1 mAb by FACS. Once verified, it would be important to add FACS results with this mAb in Figs 1A-C to have single-cell and quantitative data as well.

(4) Fig 1E-H. The authors assume that this immunization protocol induces anergic cells, but they provide no experimental evidence for this. It would be useful to show that T cells are indeed anergic in this model, especially those that are OVA-specific. The lack of IL-2 production by Cltr cells could be explained by the presence of fewer OVA-specific cells, rather than by an anergic status.

(5) Fig 2A-C and Fig 3. The use of iTregs to try to understand what is happening in vivo is problematic. iTregs are cells that have probably no equivalent in vivo, and so may have no physiological relevance. In any case, they are different from pTreg cells generated in vivo. Working with pTreg may be challenging, that is why I would suggest to generate data with purified nTreg.

(6) Fig 2D-L. The model is designed to study the role of Nrn1 in nTreg. However, the % of Foxp3+ among CD45.2 nTreg cells fell to 5-15% of CD4+ cells (Fig 2F). Since we do not know what is the % of Foxp3 among the injected cells, we do not know whether this very low % is due to very high Treg instability or to preferential expansion of contaminating Tconvs. It is possible that the % of Tconv contaminant is high since Treg were sorted using beads and not FACS on some experiments. As it is very likely that there are Tconv contaminants that would be Nrn1-/- in the group transferred with Nrn1-/- "nTreg", the higher tumor rejection could be due to an overactivation of Nrn1-/- Tconvs (rather than a defect in Nrn1-/- Treg function).

---

## [Referee Report · Reviewer #2 (Public review)]

Summary:

This manuscript explores the role of Nrn1 in T cell tolerance. A previous study has demonstrated that Nrn1 is up-regulated in the Tfr fraction of Foxp3+ T regulatory cells. These authors now confirm expression of Nrn1 in iTregs as well as report here that Nrn1 is also greatly over-expressed in anergic CD4 T cells, and this is the stepping off point for this investigation.

Most remarkably, experiments show that anergy induction is defective when T cells cannot express Nrn1. Furthermore, differentiation to a Foxp3+ iTreg phenotype is inhibited in the absence of Nrn1, and the iTregs that do develop appear functionally defective. On the other hand, the differentiation and expansion of Teff cells appears to be enhanced following deletion of Nrn1. With such defects in anergy induction as well as dysregulated Treg and Teff cell survival and function, auto reactive effector T cell activation becomes unrestrained and Nrn1-/- mice are more susceptible to severe EAE development.

Strengths:

The characterizations of T cell Nrn1 expression both in vitro and in vivo are comprehensive and convincing. The author's use of both Nrn1-/- T cells as well as anti-Nrn1 neutralizing Ab to achieve similar results is a strength. The in vivo functional studies of anergy development, Treg suppression, and EAE development are also well performed and strengthen the notion that Nrn1 is an important regulator of CD4 responsiveness.

Weaknesses:

The major weakness of this study stems from a lack of a clear molecular mechanism involving Nrn1. Previous studies of Nrn1 have suggested its role as a soluble molecule involved in intracellular communication, perhaps influencing cellular ion channel function and/or triggering downstream NFAT and mTOR activation. However, a unique receptor for Nrn1 has not been discovered and it remains unclear whether it acts in a cell-intrinsic or cell-extrinsic fashion for any particular cell type.

Data shown here provide evidence for alterations in the electrical and metabolic state of iTreg and Teff cells when the Nrn1 gene is deleted. Nrn1-/- Tregs and Teff cells each express a unique pattern of genes associated with Neurotransmitter receptor, Metal ion transmembrane transport, Amino acid transport, and mTORC1 signaling activities, different than that seen in wild-type mice. It remains unclear how Nrn1 reinforces the membrane potential and facilitates aerobic glycolysis during and after iTreg differentiation, and yet suppresses the membrane potential and restrains aerobic glycolysis during Teff cell differentiation. Importantly, naive cells lacking Nrn1 expression show normal electrical and metabolic behaviors.

---

## [Author Response]

The following is the authors’ response to the current reviews.

We thank you for sending our manuscript for the second round of review. We are encouraged by the comments from reviewer #2 that our supplementary work on naïve T cells and antibody blockade work satisfied their previous concerns and is important for our work.

The Editors raised concerns that we have shared preliminary data on Nrn1 and AMPAR double knockout mice. We apologize for our enthusiasm for these studies. Because of the publication model by eLife, we shared that data not because we needed to persuade the reviewer for publication purposes but rather to agree with the reviewer that the molecular target of Nrn1 is important, and we are progressing in understanding this subject.

The following is the authors’ response to the original reviews.

**To Reviewer #1:**

Thank you for your thorough review and comments on our work, which you described as “the role of neuritin in T cell biology studied here is new and interesting.”. We have summarized your comments into two categories: biology and investigation approach, experimental rigor, and data presentation.

Biology and Investigation approach comments:(1) Questions regarding the T cell anergy model:Major point “(4) Figure 1E-H. The authors assume that this immunization protocol induces anergic cells, but they provide no experimental evidence for this. It would be useful to show that T cells are indeed anergic in this model, especially those that are OVA-specific. The lack of IL-2 production by Cltr cells could be explained by the presence of fewer OVA-specific cells, rather than by an anergic status.”

T cell anergy is a well-established concept first described by Schwartz’s group. It refers to the hyporesponsive T cell functional state in antigen-experienced CD4 T cells (Chappert and Schwartz, 2010; Fathman and Lineberry, 2007; Jenkins and Schwartz, 1987; Quill and Schwartz, 1987). Anergic T cells are characterized by their inability to expand and to produce IL2 upon subsequent antigen re-challenge. In this paper, we have borrowed the existing in vivo T cell anergy induction model used by Mueller’s group for T cell anergy induction (Vanasek et al., 2006). Specifically, Thy1.1+ Ctrl or Nrn1-/- TCR transgenic OTII cells were co-transferred with the congenically marked Thy1.2+ WT polyclonal Treg cells into TCRα-/- mice. After anergy induction, the congenically marked TCR transgenic T cells were recovered by sorting based on Thy1.1+ congenic marker, and subsequently re-stimulation ex vivo with OVA323-339 peptide. We evaluated the T cell anergic state based on OTII cell expansion in vivo and IL2 production upon OVA323-339 restimulation ex vivo.

“The authors assume that this immunization protocol induces anergic cells, but they provide no experimental evidence for this.”

Because the anergy model by Mueller's group is well established (Vanasek et al., 2006), we did not feel that additional effort was required to validate this model as the reviewer suggested. Moreover, the limited IL2 production among the control cells upon restimulation confirms the validity of this model.

“The lack of IL-2 production by Cltr cells could be explained by the presence of fewer OVAspecific cells, rather than by an anergic status”.

Cells from Ctrl and Nrn1-/- mice on a homogeneous TCR transgenic (OTII) background were used in these experiments. The possibility that substantial variability of TCR expression or different expression levels of the transgenic TCR could have impacted IL2 production rather than anergy induction is unlikely.

Overall, we used this in vivo anergy model to evaluate the Nrn1-/- T cell functional state in comparison to Ctrl cells under the anergy induction condition following the evaluation of Nrn1 expression, particularly in anergic T cells. Through studies using this anergy model, we observed a significant change in Treg induction among OTII cells. We decided to pursue the role of Nrn1 in Treg cell development and function rather than the biology of T cell anergy as evidenced by subsequent experiments.

Minor points “(6) On which markers are anergic cells sorted for RNAseq analysis?”

Cells were sorted out based on their congenic marker marking Ctrl or Nrn1-/- OTII cells transferred into the host mice. We did not specifically isolate anergic cells for sequencing.

(2) Question regarding the validity of iTreg differentiation model.Major point: “(5) Figure 2A-C and Figure 3. The use of iTregs to try to understand what is happening in vivo is problematic. iTregs are cells that have probably no equivalent in vivo, and so may have no physiological relevance. In any case, they are different from pTreg cells generated in vivo. Working with pTreg may be challenging, that is why I would suggest generating data with purified nTreg. Moreover, it was shown in the article of Gonzalez-Figueroa 2021 that Nrn1-/- nTreg retained a normal suppressive function, which would not be what is concluded by the authors of this manuscript. Moreover, we do not even know what the % of Foxp3 cells is in the iTreg used (after differentiation and 20h of re-stimulation) and whether this % is the same between Ctlr and Nrn1 KO cells.”.

We thank Reviewer #1 for their feedback. While it is true that iTregs made in vitro and in vivo generated pTregs display several distinctions (e. g., differences in Foxp3 expression stability, for example), we strongly disagree with this statement by Revieweer#1 “The use of iTregs to try to understand what is happening in vivo is problematic. iTregs are cells that have probably no equivalent in vivo, and so may have no physiological relevance.” The induced Treg cell (iTreg) model was established over 20 years ago (Chen et al., 2003; Zheng et al., 2002), and the model is widely adopted with over 2000 citations. Further, it has been instrumental in understanding different aspects of regulatory T cell biology (Hurrell et al., 2022; John et al., 2022; Schmitt and Williams, 2013; Sugiura et al., 2022).

Because we have observed reduced pTreg generation in vivo, we choose to use the in vitro iTreg model system to understand the mechanistic changes involved in Treg cell differentiation and function, specifically, neuritin’s role in this process. We have made no claim that iTreg cell biology is identical to pTreg generated in vivo or nTreg cells. However, the iTreg culture system has proved to be a good in vitro system for deciphering molecular events involved in complex processes. As such, it remains a commonly used approach by many research groups in the Treg cell field (Hurrell et al., 2022; John et al., 2022; Sugiura et al., 2022). Moreover, applying the iTreg in vitro culture system has been instrumental in helping us identify the cell electrical state change in Nrn1-/- CD4 cells and revealed the biological link between Nrn1 and the ionotropic AMPA receptor (AMPAR), which we will discuss in the subsequent discussion. It is technically challenging to use nTreg cells for T cell electrical state studies due to their heterogeneous nature from development in an in vivo environment and the effect of manipulation during the nTreg cell isolation process, which can both affect the T cell electrical state.

“Moreover, it was shown in the article of Gonzalez-Figueroa 2021 that Nrn1-/- nTreg retained a normal suppressive function, which would not be what is concluded by the authors of this manuscript.”

We have also carried out nTreg studies in vitro in addition to iTreg cells. Similar to Gonzalez-Figueroa et al.'s findings, we did not observe differences in suppression function between Nrn1-/- and WT nTreg using the in vitro suppression assay. However, Nrn1-/- nTreg cells revealed reduced suppression function in vivo (Fig. 2D-L). In fact, Gonzalez-Figueroa et al. observed reduced plasma cell formation after OVA immunization in Treg-specific Nrn1-/- mice, implicating reduced suppression from Nrn1-/- follicular regulatory T (Tfr) cells. Thus, our observation of the reduced suppression function of Nrn1-/- nTreg toward effector T cell expansion, as presented in Fig. 2D-L, does not contradict the results from Gonzalez-Figueroa et al. Rather, the conclusions of these two studies agree that Nrn1 can play important roles in immune suppression observable in vivo that are not captured readily by the in vitro suppression assay.

“Moreover, we do not even know what the % of Foxp3 cells is in the iTreg used (after differentiation and 20h of re-stimulation) and whether this % is the same between Ctlr and Nrn1 KO cells.”

We have stated in the manuscript on page 7 line 208 that “Similar proportions of Foxp3+ cells were observed in Nrn1-/- and Ctrl cells under the iTreg culture condition, suggesting that Nrn1 deficiency does not significantly impact Foxp3+ cell differentiation”. In the revised manuscript, we will include the data on the proportion of Foxp3+ cells before iTreg restimulation.

(3) Confirmation of transcriptomic data regarding amino acids or electrolytes transport changeMinor point“(3) Would not it be possible to perform experiments showing the ability of cells to transport amino acids or electrolytes across the plasma membrane? This would be a more interesting demonstration than transcriptomic data.”

We appreciate Review# 1’s suggestion regarding “perform experiments showing the ability of cells to transport amino acids or electrolytes across the plasma membrane”. We have indeed already performed such experiments corroborating the transcriptomics data on differential amino acid and nutrient transporter expression. Specifically, we loaded either iTreg or Th0 cells with membrane potential (MP) dye and measured MP level change after adding the complete set of amino acids (complete AA). Upon entry, the charge carried by AAs may transiently affect cell membrane potential. Different AA transporter expression patterns may show different MP change patterns upon AA entry, as we showed in Author response image 1. We observed reduced MP change in Nrn1-/- iTreg compared to the Ctrl, whereas in the context of Th0 cells, Nrn1-/- showed enhanced MP change than the Ctrl. We can certainly include these data in the revised manuscript.

**Author response image 1. sa3fig1:** Membrane potential change induced by amino acids entry. a. Nrn1-/- or WT iTreg cells loaded with MP dye and MP change was measured upon the addition of a complete set of AAs. b. Nrn1-/- or WT Th0 cells loaded with MP dye and MP change was measured upon the addition of a complete set of AAs.

(4) EAE experiment data assessmentMinor point ”(5) Figure 5F. How are cells re-stimulated? If polyclonal stimulation is used, the experiment is not interesting because the analysis is done with lymph node cells. This analysis should either be performed with cells from the CNS or with MOG restimulation with lymph node cells.”

In the EAE study, the Nrn1-/- mice exhibit similar disease onset but a protracted non-resolving disease phenotype compared to the WT control mice. Several reasons may contribute to this phenotype: 1. Enhanced T effector cell infiltration/persistence in the central nervous system (CNS); 2. Reduced Treg cell-mediated suppression to the T effector cells in the CNS; 3. Protracted non-resolving inflammation at the immunization site has the potential to continue sending T effector cells into CNS, contributing to persistent inflammation. Based on this reasoning, we examined the infiltrating T effector cell number and Treg cell proportion in the CNS. We also restimulated cells from draining lymph nodes close to the inflammation site, looking for evidence of persistent inflammation. When mice were harvested around day 16 after immunization, the inflammation at the local draining lymph node should be at the contraction stage. We stimulated cells with PMA and ionomycin intended to observe all potential T effector cells involved in the draining lymph node rather than only MOG antigen-specific cells. We disagree with Reviewer #1’s assumption that “This analysis should either be performed with cells from the CNS or with MOG restimulation with lymph node cells.”. We think the experimental approach we have taken has been appropriately tailored to the biological questions we intended to answer.

Experimental rigor and data presentation.(1) data labeling and additional supporting dataMajor points(2) The authors use Nrn1+/+ and Nrn1+/- cells indiscriminately as control cells on the basis of similar biology between Nrn1+/+ and Nrn1+/- cells at homeostasis. However, it is quite possible that the Nrn1+/- cells have a phenotype in situations of in vitro activation or in vivo inflammation (cancer, EAE). It would be important to discriminate Nrn1+/- and Nrn1+/+ cells in the data or to show that both cell types have the same phenotype in these conditions too.(3) Figure 1A-D. Since the authors are using the Nrp1 KO mice, it would be important to confirm the specificity of the anti-Nrn1 mAb by FACS. Once verified, it would be important to add FACS results with this mAb in Figures 1A-C to have single-cell and quantitative data as well.Minor points(1) Line 119, 120 of the text. It is said that one of the most up-regulated genes in anergic cells is Nrn1 but the data is not shown.(2) For all figures showing %, the titles of the Y axes are written in an odd way. For example, it is written "Foxp3% CD4". It would be more conventional and clearer to write "% Foxp3+ / CD4+" or "% Foxp3+ among CD4+".(4) For certain staining (Figure 3E, H) it would be important to show the raw data, in addition to MFI or % values.

We can adapt the labeling and provide additional data, including Nrn1 staining on Treg cells and flow graphs for pmTOR and pS6 staining (Fig. 3H), as requested by Reviewer #1.

(2) Experimental rigor:General comments:“However, it is disappointing that reading this manuscript leaves an impression of incomplete work done too quickly.”

We were discouraged to receive the comment, “this manuscript leaves an impression of incomplete work done too quickly.” Our study of this novel molecule began without any existing biological tools such as antibodies, knockout mice, etc. Over the past several years, we have established our own antibodies for Nrn1 detection, obtained and characterized Nrn1 knockout mice, and utilized multiple approaches to identify the molecular mechanism of Nrn1 function. Through the use of the in vitro iTreg system described in this manuscript, we identified the association of Nrn1 deficiency with cell electrical state change, potentially connected to AMPAR function. We have further corroborated our findings by generating Nrn1 and AMPAR T cell specific double knockout mice and confirmed that T cell specific AMPAR deletion could abrogate the phenotype caused by the Nrn1 deficiency (see Support Figure 2). We did not include the double knockout data in the current manuscript because AMPAR function has not yet been studied thoroughly in T cell biology, and we feel this topic warrants examination in its own right. However, the unpublished data support the finding that Nrn1 modulates the T cell electrical state and, consequently, metabolism, ultimately influencing tolerance and immunity. In its current form, the manuscript represents the first characterization of the novel molecule Nrn1 in anergic cells, Tregs, and effector T cells. While this work has led to several exciting additional questions, we disagree that the novel characterization we have presented Is incomplete. We feel that our present data set, which squarely highlights Nrn1’s role as an important immune regulator while shedding unprecedented light on the molecular events involved, will be of considerable interest to a broad field of researchers.

“Multiple models have been used, but none has been studied thoroughly enough to provide really conclusive and unambiguous data. For example, 5 different models were used to study T cells in vivo. It would have been preferable to use fewer, but to go further in the study of mechanisms.”

We have indeed used multiple in vivo models to reveal Nrn1's function in Treg differentiation, Treg suppression function, T effector cell differentiation and function, and the overall impact on autoimmune disease. Because the impact of ion channel function is often context-dependent, we examined the biological outcome of Nrn1 deficiency in several in vivo contexts. We would appreciate it if Reviewer#1 would provide a specific example, given the Nrn1 phenotype, of how to proceed deeper to investigate the electrical change in the in vivo models.

“Major points(1) A real weakness of this work is the fact that in most of the results shown, there are few biological replicates with differences that are often small between Ctrl and Nrn1 -/-. The systematic use of student's t-test may lead to thinking that the differences are significant, which is often misleading given the small number of samples, which makes it impossible to know whether the distributions are Gaussian and whether a parametric test can be used. RNAseq bulk data are based on biological duplicates, which is open to criticism.”

We respectfully disagree with Reviewer #1 on the question of statistical power and significance to our work. We have used 5-8 mice/group for each in vivo model and 3-4 technical replicates for the in vitro studies, with a minimum of 2-3 replicate experiments. These group sizes and replication numbers are in line with those seen in high-impact publications. While some differences between Ctrl and Nrn1-/- appear small, they have significant biological consequences, as evidenced by the various Nrn1-/- in vivo phenotypes. Furthermore, we believe we have subjected our data to the appropriate statistical tests to ensure rigorous analysis and representation of our findings.

**To Reviewer #2.**

We thank Reviewer #2 for the careful review of the manuscript. We especially appreciate the comments that “The characterizations of T cell Nrn1 expression both in vitro and in vivo are comprehensive and convincing. The in vivo functional studies of anergy development, Treg suppression, and EAE development are also well done to strengthen the notion that Nrn1 is an important regulator of CD4 responsiveness.”

“The major weakness of this study stems from a lack of a clear molecular mechanism involving Nrn1. “

We fully understand this comment from Reviewer #2. The main mechanism we identified contributing to the functional defect of Nrn1-/- T cells involves novel effects on the electric and metabolic state of the cells. Although we referenced neuronal studies that indicate Nrn1 is the auxiliary protein for the ionotropic AMPA-type glutamate receptor (AMPAR) and may affect AMPAR function, we did not provide any evidence in this manuscript as the topic requires further in-depth study.

For the benefit of this discussion, we include our preliminary Nrn1 and AMPAR double knockout data (Author response image 2), which indicates that abrogating AMPAR expression can compensate for the defect caused by Nrn1 deficiency in vitro and in vivo. This preliminary data supports the notion that Nrn1 modulates AMPAR function, which causes changes in T cell electric and metabolic state, influencing T cell differentiation and function.

Author response image 2.

Deletion of AMPAR expression in T cells compensates for the defect caused by

Nrn1 deficiency. Nrn1-/- mice were crossed with T cell-specific AMPAR knockout mice (AMPARfl/flCD4Cre+) mice. The following mice were generated and used in the experiment: T cell specific AMPAR-knockout and Nrn1 knockout mice (AKONKO), Nrn1 knockout mice (AWTNKO), Ctrl mice (AWTNWT). a. Deletion of AMPAR compensates for the iTreg cell defect observed in Nrn1-/- CD4 cells. iTreg live cell proportion, cell number, and Ki67 expression among Foxp3+ cells 3 days after aCD3 restimulation. b. Deletion of AMPAR in T cells abrogates the enhanced autoimmune response in Nrn1-/- Mouse in the EAE disease model. Mouse relative weight change and disease score progression after EAE disease induction.

Ion channels can influence cell metabolism through multiple means (Vaeth and Feske, 2018; Wang et al., 2020). First, ion channels are involved in maintaining cell resting membrane potential. This electrical potential difference across the cell membrane is essential for various cellular processes, including metabolism (Abdul Kadir et al., 2018; Blackiston et al., 2009; Nagy et al., 2018; Yu et al., 2022). Second, ion channels facilitate the movement of ions across cell membranes. These ions are essential for various metabolic processes. For example, ions like calcium (Ca2+), potassium (K+), and sodium (Na+) play crucial roles in signaling pathways that regulate metabolism (Kahlfuss et al., 2020). Third, ion channel activity can influence cellular energy balance due to ATP consumption associated with ion transport to maintain ion balances (Erecińska and Dagani, 1990; Gerkau et al., 2019). This, in turn, can impact processes like ATP production, which is central to cellular metabolism. Thus, ion channel expression and function determine the cell’s bioelectric state and contribute to cell metabolism (Levin, 2021).

Because the AMPAR function has not been thoroughly studied using a genetic approach in T cells, we do not intend to include the double knockout data in this manuscript before fully characterizing the T cell-specific AMPAR knockout mice.

“Although the biochemical and informatics studies are well-performed, it is my opinion that these results are inconclusive in part due to the absence of key "naive" control groups. This limits my ability to understand the significance of these data.

Specifically, studies of the electrical and metabolic state of Nrn1-/- inducible Treg cells (iTregs) would benefit from similar data collected from wild-type and Nrn1-/- naive CD4 T cells.”

We appreciate the reviewer’s comments. This comment reflects two concerns in data interpretation:

(1) Are Nrn1-/- naïve T cells fundamentally different from WT cells? Does this fundamental difference contribute to the observed electrical and metabolic phenotype in iTreg or Th0 cells? This is a very good question we will perform the experiments as the reviewer suggested. While Nrn1 is expressed at a basal (low) level in naïve T cells, deletion of Nrn1 may cause changes in naïve T cell phenotype.

(2) Is the Nrn1-/- phenotype caused by Nrn1 functional deficiency or due to the secondary effect of Nrn1 deletion, such as non-physiological cell membrane structure changes?

We have done the following experiment to address this concern. We have cultured WT T cells in the presence of Nrn1 antibody and compared the outcome with Nrn1-/- iTreg cells (Figure 3-figure supplement 2D,E,F). WT iTreg cells under antibody blockade exhibited similar changes as Nrn1-/- iTreg cells, confirming the physiological relevance of the Nrn1-/- phenotype.

Manuscript Revision based on the Reviewer’s suggestions:

**Reviewer #1:**
Major points (3) Figure 1A-D. Since the authors are using the Nrp1 KO mice, it would be important to confirm the specificity of the anti-Nrn1 mAb by FACS.

Following the suggestion by Reviewer#1, We have included the Nrn1 Ab staining on activated Nrn1-/- CD4 cells in Figure 1D. We have also added the staining of cell surface Nrn1 on Treg cells in Figure 1-figure supplement 1D.

Major point: (5) “Moreover, we do not even know what the % of Foxp3 cells is in the iTreg used (after differentiation and 20h of re-stimulation) and whether this % is the same between Ctlr and Nrn1 KO cells.”

In the revised manuscript, we have included the proportion of Foxp3+ cells among Nrn1-/- and ctrl iTreg cells developed under the iTreg culture condition in Figure 2A.

Minor points(2) For all figures showing %, the titles of the Y axes are written in an odd way. For example, it is written "Foxp3% CD4". It would be more conventional and clearer to write "% Foxp3+ / CD4+" or "% Foxp3+ among CD4+".

Following reviewer#1’s suggestion, we have changed the Y-axis label in all the relevant figures.

(3) Would not it be possible to perform experiments showing the ability of cells to transport amino acids or electrolytes across the plasma membrane? This would be a more interesting demonstration than transcriptomic data.”

We appreciate Review# 1’s suggestion regarding “perform experiments showing the ability of cells to transport amino acids or electrolytes across the plasma membrane”. We have used AAinduced cellular MP changes to confirm differential AA transporter expression patterns and their impact on cellular MP levels. The data are included in the revised manuscript in Figure 3H and Figure 4K.

(4) For certain staining (Figure 3E, H) it would be important to show the raw data, in addition to MFI or % values.

We appreciated Reviewer #1’s suggestion and have included the histogram staining data for Figure 3E. We have moved the original Figure 3H to the supplemental figure and included the histogram staining data in Figure 3-figure supplement 1C. Similarly, we have included the histogram staining data in Figure 4-figure supplement 1C.

**Reviewer#2:**
“Although the biochemical and informatics studies are well-performed, it is my opinion that these results are inconclusive in part due to the absence of key "naive" control groups. This limits my ability to understand the significance of these data.Specifically, studies of the electrical and metabolic state of Nrn1-/- inducible Treg cells (iTregs) would benefit from similar data collected from wild-type and Nrn1-/- naive CD4 T cells.”

We greatly appreciate Reviewer#2’s suggestion and have carried out experiments on naïve CD4 cells derived from Nrn1-/- and WT mice. We have compared membrane potential, AA-induced MP change between Nrn1-/- and WT naïve T cells, and the metabolic state of Nrn1-/- and WT naïve T cells by carrying out glucose stress tests and mitochondria stress tests using a seahorse assay. Moreover, to investigate whether the phenotype revealed in Nrn1-/- CD4 cells was caused by a secondary effect of cell membrane structure change due to Nrn1 deletion, we carried out Nrn1 antibody blockade in WT CD4 cells and investigated the phenotypic change. These new results are included in Figure 3-figure supplement 2.

Reference:

Abdul Kadir, L., M. Stacey, and R. Barrett-Jolley. 2018. Emerging Roles of the Membrane Potential: Action Beyond the Action Potential. Front Physiol 9:1661.

Blackiston, D.J., K.A. McLaughlin, and M. Levin. 2009. Bioelectric controls of cell proliferation: ion channels, membrane voltage and the cell cycle. Cell Cycle 8:3527-3536.

Chappert, P., and R.H. Schwartz. 2010. Induction of T cell anergy: integration of environmental cues and infectious tolerance. Current opinion in immunology 22:552-559.

Chen, W., W. Jin, N. Hardegen, K.J. Lei, L. Li, N. Marinos, G. McGrady, and S.M. Wahl. 2003. Conversion of peripheral CD4+CD25- naive T cells to CD4+CD25+ regulatory T cells by TGF-beta induction of transcription factor Foxp3. The Journal of experimental medicine 198:1875-1886.

Erecińska, M., and F. Dagani. 1990. Relationships between the neuronal sodium/potassium pump and energy metabolism. Effects of K+, Na+, and adenosine triphosphate in isolated brain synaptosomes. J Gen Physiol 95:591-616.

Fathman, C.G., and N.B. Lineberry. 2007. Molecular mechanisms of CD4+ T-cell anergy. Nat Rev Immunol 7:599-609.

Gerkau, N.J., R. Lerchundi, J.S.E. Nelson, M. Lantermann, J. Meyer, J. Hirrlinger, and C.R. Rose. 2019. Relation between activity-induced intracellular sodium transients and ATP dynamics in mouse hippocampal neurons. The Journal of physiology 597:5687-5705.

Hurrell, B.P., D.G. Helou, E. Howard, J.D. Painter, P. Shafiei-Jahani, A.H. Sharpe, and O. Akbari. 2022. PD-L2 controls peripherally induced regulatory T cells by maintaining metabolic activity and Foxp3 stability. Nature communications 13:5118.

Jenkins, M.K., and R.H. Schwartz. 1987. Antigen presentation by chemically modified splenocytes induces antigen-specific T cell unresponsiveness in vitro and in vivo. The Journal of experimental medicine 165:302-319.

John, P., M.C. Pulanco, P.M. Galbo, Jr., Y. Wei, K.C. Ohaegbulam, D. Zheng, and X. Zang. 2022. The immune checkpoint B7x expands tumor-infiltrating Tregs and promotes resistance to anti-CTLA-4 therapy. Nature communications 13:2506.

Kahlfuss, S., U. Kaufmann, A.R. Concepcion, L. Noyer, D. Raphael, M. Vaeth, J. Yang, P. Pancholi, M. Maus, J. Muller, L. Kozhaya, A. Khodadadi-Jamayran, Z. Sun, P. Shaw, D. Unutmaz, P.B. Stathopulos, C. Feist, S.B. Cameron, S.E. Turvey, and S. Feske. 2020. STIM1-mediated calcium influx controls antifungal immunity and the metabolic function of nonpathogenic Th17 cells. EMBO molecular medicine 12:e11592.

Levin, M. 2021. Bioelectric signaling: Reprogrammable circuits underlying embryogenesis, regeneration, and cancer. Cell 184:1971-1989.

Nagy, E., G. Mocsar, V. Sebestyen, J. Volko, F. Papp, K. Toth, S. Damjanovich, G. Panyi, T.A. Waldmann, A. Bodnar, and G. Vamosi. 2018. Membrane Potential Distinctly Modulates Mobility and Signaling of IL-2 and IL-15 Receptors in T Cells. Biophys J 114:2473-2482.

Quill, H., and R.H. Schwartz. 1987. Stimulation of normal inducer T cell clones with antigen presented by purified Ia molecules in planar lipid membranes: specific induction of a long-lived state of proliferative nonresponsiveness. Journal of immunology (Baltimore, Md. : 1950) 138:3704-3712.

Schmitt, E.G., and C.B. Williams. 2013. Generation and function of induced regulatory T cells. Frontiers in immunology 4:152.

Sugiura, A., G. Andrejeva, K. Voss, D.R. Heintzman, X. Xu, M.Z. Madden, X. Ye, K.L. Beier, N.U. Chowdhury, M.M. Wolf, A.C. Young, D.L. Greenwood, A.E. Sewell, S.K. Shahi, S.N. Freedman, A.M. Cameron, P. Foerch, T. Bourne, J.C. Garcia-Canaveras, J. Karijolich, D.C. Newcomb, A.K. Mangalam, J.D. Rabinowitz, and J.C. Rathmell. 2022. MTHFD2 is a metabolic checkpoint controlling effector and regulatory T cell fate and function. Immunity 55:65-81.e69.

Vaeth, M., and S. Feske. 2018. Ion channelopathies of the immune system. Current opinion in immunology 52:39-50.

Vanasek, T.L., S.L. Nandiwada, M.K. Jenkins, and D.L. Mueller. 2006. CD25+Foxp3+ regulatory T cells facilitate CD4+ T cell clonal anergy induction during the recovery from lymphopenia. Journal of immunology (Baltimore, Md. : 1950) 176:5880-5889.

Wang, Y., A. Tao, M. Vaeth, and S. Feske. 2020. Calcium regulation of T cell metabolism. Current opinion in physiology 17:207-223.

Yu, W., Z. Wang, X. Yu, Y. Zhao, Z. Xie, K. Zhang, Z. Chi, S. Chen, T. Xu, D. Jiang, X. Guo, M. Li, J. Zhang, H. Fang, D. Yang, Y. Guo, X. Yang, X. Zhang, Y. Wu, W. Yang, and D. Wang. 2022. Kir2.1-mediated membrane potential promotes nutrient acquisition and inflammation through regulation of nutrient transporters. Nature communications 13:3544.

Zheng, S.G., J.D. Gray, K. Ohtsuka, S. Yamagiwa, and D.A. Horwitz. 2002. Generation ex vivo of TGF-beta-producing regulatory T cells from CD4+CD25- precursors. Journal of immunology (Baltimore, Md. : 1950) 169:4183-4189.